# Neural correlates of ingroup bias for prosociality in rats

Inbal Ben-Ami Bartal[1,2,3,4]*, Jocelyn M Breton[2,4†], Huanjie Sheng[2‡], Kimberly LP Long[2,4§], Stella Chen[2,4], Aline Halliday[2], Justin W Kenney[5#], Anne L Wheeler[5,6], Paul Frankland[5,6,7], Carrie Shilyansky[8], Karl Deisseroth[9,10,11], Dacher Keltner[4,12], Daniela Kaufer[2,4,7]

[1]School of Psychological Sciences, Tel-Aviv University, Tel Aviv, Israel; [2]Department of Integrative Biology, University of California, Berkeley, Berkeley, United States; [3]Sagol School of Neuroscience, Tel-Aviv University, Tel Aviv, Israel; [4]Helen Wills Neuroscience Institute, University of California, Berkeley, Berkeley, United States; [5]The Hospital for Sick Children, Toronto, Neuroscience and Mental Health Program, Toronto, Canada; [6]Physiology Department, University of Toronto, Toronto, Canada; [7]Canadian Institute for Advanced Research, Toronto, Canada; [8]Department of Psychiatry and Behavioral Sciences, Stanford University, Stanford, United States; [9]Department of Bioengineering, Stanford University, Stanford, United States; [10]Department of Psychiatry, Stanford University, Stanford, United States; [11]Howard Hughes Medical Institute, Stanford University, Stanford, United States; [12]Department of Psychology, University of California, Berkeley, Berkeley, United States

*For correspondence: inbalbe@tauex.tau.ac.il

Present address: [†]Psychiatry, Columbia University, New York, United States; [‡] Roche Sequencing Solutions, Inc, Santa Clara, United States; [§]Psychiatry, University of California, San-Francisco, San-Francisco, United States; [#]Department of Biological Sciences, Wayne State University, Detroit, United States

Competing interests: The authors declare that no competing interests exist.

**Abstract** Prosocial behavior, in particular helping others in need, occurs preferentially in response to distress of one's own group members. In order to explore the neural mechanisms promoting mammalian helping behavior, a discovery-based approach was used here to identify brain-wide activity correlated with helping behavior in rats. Demonstrating social selectivity, rats helped others of their strain ('ingroup'), but not rats of an unfamiliar strain ('outgroup'), by releasing them from a restrainer. Analysis of brain-wide neural activity via quantification of the early-immediate gene c-Fos identified a shared network, including frontal and insular cortices, that was active in the helping test irrespective of group membership. In contrast, the striatum was selectively active for ingroup members, and activity in the nucleus accumbens, a central network hub, correlated with helping. In vivo calcium imaging showed accumbens activity when rats approached a trapped ingroup member, and retrograde tracing identified a subpopulation of accumbens-projecting cells that was correlated with helping. These findings demonstrate that motivation and reward networks are associated with helping an ingroup member and provide the first description of neural correlates of ingroup bias in rodents.

## Introduction

Humans are an intensely social species, with complex social interactions and an ability to know and share others' emotional states (**Wilson, 2012**). We often behave prosocially, acting with the intention of benefiting others or improving their well-being (**Cronin, 2012**). Yet, prosocial behavior tends to be extended preferentially between group members and is less likely to be offered to others outside the group (**Eisenberg et al., 2010**). Reduced prosocial motivation towards outgroup members poses a major challenge for a diverse society, where members of different groups, such as racial and religious ones, coexist (**Dovidio, 2010**). Yet overcoming biases can be difficult, even for highly

motivated individuals (*Fiske, 2002*), and multiple strategies for bias reduction have proved only partly successful in the long term (*Paluck et al., 2021*). Empathy, the ability to perceive and share the emotions of others, coupled with a motivation to care for their well-being (*Batson, 2009*; *Decety et al., 2016*), is considered to be a major driver for prosocial behavior in humans (*Batson, 2011*; *Eisenberg, 2007*). An "empathy gap" towards outgroup members is well documented in humans (*Cheon et al., 2011*; *Chiao et al., 2008*; *Cikara et al., 2011*; *Eisenberg, 1989*; *Gutsell and Inzlicht, 2012*) and is thought to lie at the root of reduced prosocial motivation towards other groups (*Amodio et al., 2004*; *Echols and Correll, 2012*; *Levine et al., 2005*; *Stürmer et al., 2006*). In line with this idea, humans are willing to pay a greater personal cost to prevent pain to ingroup members (*Hein et al., 2010*) and display a dampened neural response to outgroup members' pain (*Avenanti et al., 2010*; *Ruckmann et al., 2015*; *Xu et al., 2009*).

While the extent and complexity of human prosocial behavior is unique, basic empathic responses and prosocial behavior are common in social species across the phylogenetic spectrum (*Brosnan, 2020*), and rely on evolutionarily conserved biological mechanisms (*Decety et al., 2012*). Empathy is thought to have evolved in the context of parental care and subsequently integrated into the natural behavioral repertoire of social species (*de Waal, 2008*; *Preston and de Waal, 2002*). As in humans, other animals, including rodents, determine behavior towards others in large part according to social identity, with prosocial actions typically extended towards ingroup members rather than unaffiliated others (*Anacker and Beery, 2013*; *Campbell and de Waal, 2011*; *Fu et al., 2012*; *Hamilton, 1964*; *Mahajan et al., 2011*; *Nowak et al., 2010*). The terms 'ingroup' and 'outgroup,' while often used in regard to cultural processes underlying social identity in humans, are also used for describing socially selective affiliation in non-human animals (*Masuda and Fu, 2015*; *Nakamura and Masuda, 2012*; *Robinson and Barker, 2017*), and are adopted here, despite likely differences in their neurobiological structure across species.

In recent years, evidence has shown that rodents experience stress and fear in response to observing others in distress (*Cox and Reichel, 2021*; *Hernandez-Lallement et al., 2020*; *Meyza et al., 2017*), console distressed mates (*Burkett et al., 2016*), and act for the benefit of others (*Ben-Ami Bartal et al., 2011*; *Cox and Reichel, 2020*; *Hernandez-Lallement et al., 2014*; *Márquez et al., 2015*; *Rice and Gainer, 1962*; *Sato et al., 2015*). These studies provide robust evidence that rats, a highly social species, are moved to action by others' distress and strive to help them, thereby demonstrating the basic components of empathic helping.

In the 'helping behavior test' (HBT) used below, it has been previously demonstrated that rats help conspecifics by releasing them from a restrainer, without any prior training or reward and even in the absence of social contact after helping (*Ben-Ami Bartal et al., 2011*). Importantly, rats demonstrate an ingroup bias, releasing 'ingroup members' (rats of the same strain, whether familiar or not), but not 'outgroup members' (rats of an unfamiliar strain) (*Ben-Ami Bartal et al., 2014*).

Here we employed a discovery-driven approach to compare brain-wide activation patterns in response to trapped ingroup and outgroup members following the HBT. This investigation, which aimed at providing a broad and unbiased overview, led to the identification of central hubs specifically active during the ingroup condition, where rats demonstrated prosocial intent.

Understanding the neural mechanisms at the root of these phenomena is imperative for the advancement of novel interventions aimed at eliminating social bias. We seek to understand how the brain represents group membership categories, and how this information determines the cascade of evaluative and affective responses to others' distress, ultimately influencing the motivation to approach and help others in need. Rodents provide an ideal model for exploring neural activity in this context. Thus, here we set out to describe the neural activity associated with rats' ingroup bias for prosocial behavior.

## Results

### Rats demonstrate an ingroup bias for prosocial behavior

In the first set of experiments, male Sprague–Dawley (SD) rats were tested for helping behavior with a trapped cagemate of the same strain ('HBT ingroup' condition, n=8) or a stranger of the black-caped Long–Evans strain ('HBT outgroup' condition, n=8; *Figure 1A*). Along the 2 weeks of testing, most rats in the HBT ingroup condition learned to open the restrainer and consistently released the

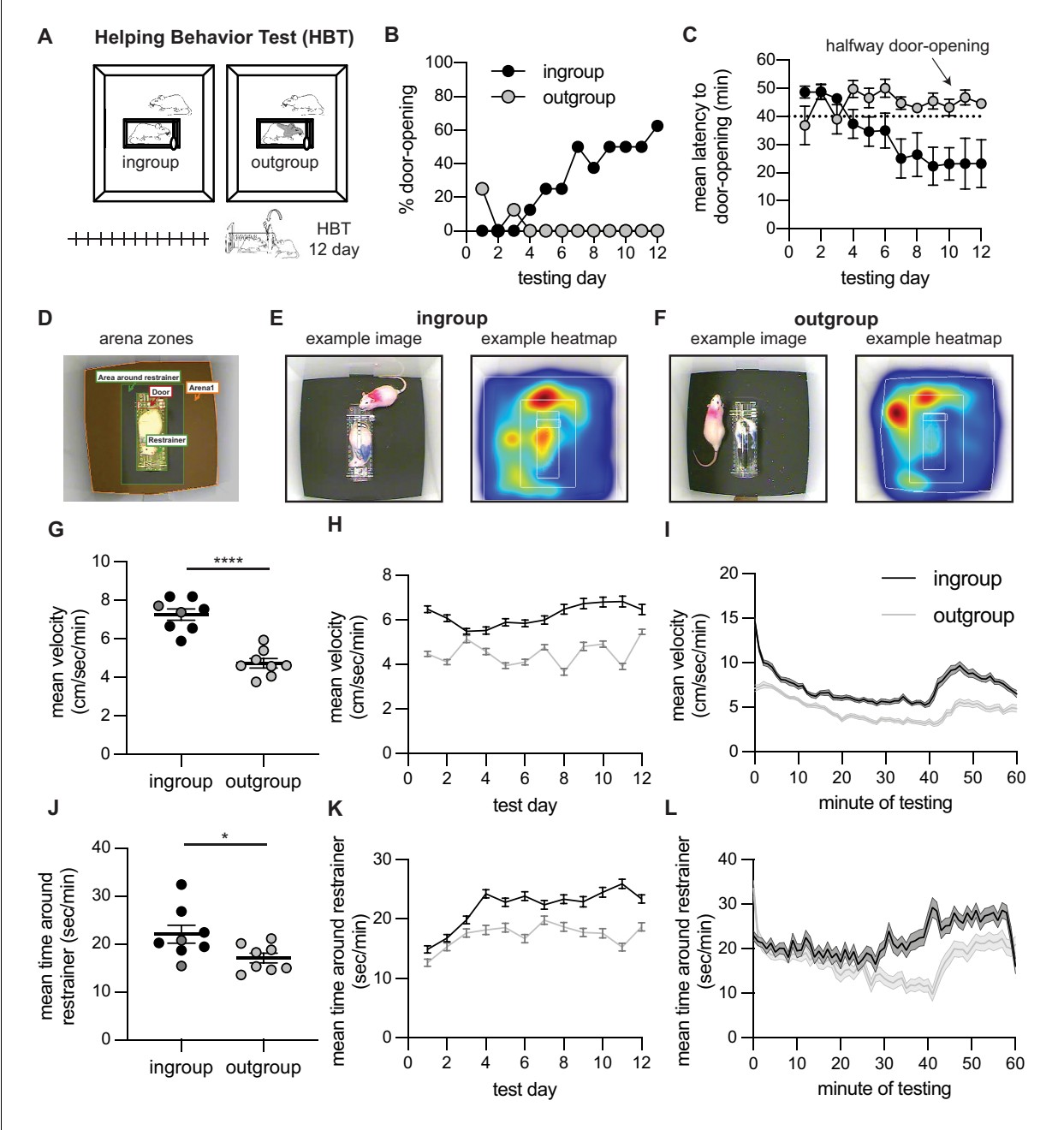

**Figure 1.** Helping behavior for adult rats tested with ingroup and outgroup members. Adult rats selectively helped ingroup members. (A) Diagram of the helping behavior test (HBT) with a trapped rat. The trapped rat was either a cagemate of the same strain (ingroup, left) or a stranger of an unfamiliar strain (outgroup, right). (B, C) Rats released ingroup members but not outgroup members, as expressed by % door-openings (B) and mean (± SEM) latency to open (C) across testing sessions. The dashed line indicates the half-way door-opening by the experimenter. (D–F) Representative movement patterns of rats tested with an ingroup or outgroup member, depicted by a heatmap of the rat's location along the session. Rats were more active (G–I) and spent more time around the restrainer (J–L) in the presence of a trapped ingroup member than an outgroup member. Lighter dots indicate non-openers, dark dots are openers.

trapped rat on following sessions (n=6/8 rats became 'openers,' see Materials and methods). Along the days of testing, the percent of door-openings increased (Cochran's Q = 31.43, df = 11, p=0.001, *Figure 1B*) and the mean latency to door-opening decreased (Friedman, $X^2$ = 20.12, df = 11, p=0.043, *Figure 1C*). In contrast, and in line with prior findings, rats tested with a trapped outgroup member rarely released the trapped rat (n=0/8 became 'openers'), and door-opening did not

increase along testing sessions (Cochran's Q and Friedman, p>0.05, *Figure 1B, C*). Rat's movement patterns during testing were recorded by video and analyzed offline (*Figure 1D–F*). Across the 12 days, rats in the HBT ingroup condition were more active in total ($t_{(14)}$ = 3.8, p=0.002, *Figure 1G*), as well as across days (mixed model analysis [MMA], $F_{(1,1037)}$ = 372.3, p<0.001; Bonferroni-corrected post-hoc comparisons, p<0.05 for all days, *Figure 1H*) and minutes (MMA, $F_{(1,1304)}$ = 556.1, p<0.001; Bonferroni, p<0.05, *Figure 1I*) of testing. Rats in the ingroup condition also spent more time in the area around the restrainer ($t_{(14)}$ = 2.35, p=0.03, *Figure 1J*). This difference emerged in later days (MMA, $F_{(1,1650)}$ = 94.2, p<0.001; Bonferroni, p<0.05 for days 4–6, 8–12, *Figure 1K*) and minutes of testing (MMA, $F_{(1,1690)}$ = 103.85, p<0.001; Bonferroni, p<0.05 for minutes 30–40, *Figure 1L*). During times in the session when the restrainer door was closed, this pattern was maintained yet less pronounced both for velocity (MMA, $F_{(1,486)}$ = 172, p<0.001; Bonferroni, p<0.05, barring day 3, and $F_{(1,1140)}$ = 211.3, p<0.01, Bonferroni p < 0.05 for all minutes) and time around the restrainer (MMA, $F_{(1,1037)}$ = 11.5, p<0.01; Bonferroni p<0.05 for days 11–12, and $F_{(1,1520)}$ = 12.6, p<0.01; Bonferroni p<0.05 for minutes 30–40). In sum, rats released trapped cagemates of the same strain, but not strangers of an unfamiliar strain, demonstrating an ingroup bias for prosocial behavior.

In order to map the brain-wide activation involved in this selective prosocial response, the immediate early-gene marker c-Fos was quantified as an index of neural activity (*Guzowski et al., 2005*) across the brain of rats tested in the HBT conditions (n = 84 samples per rat, *Figure 2A–C*, *Figure 2—figure supplement 1A, B*, *Supplementary file 1*). c-Fos expression reflects neural activity during the final testing session, during which restrainers were latched shut. Thus, rats in both HBT groups were in the presence of a trapped rat for the session's entire duration and had an objectively similar experience on the c-Fos imaging day, despite their different history of door-opening. c-Fos was also quantified for several control conditions (*Figure 2—figure supplement 2A–D*). As a non-social control task, rats were tested in the HBT with a restrainer containing chocolate chips ('chocolate,' n = 8). In another condition, rats were tested with either an ingroup or outgroup member trapped in a latched, unopenable restrainer for three daily sessions in order to capture the neural response to a trapped rat without the experience of door-opening ('brief ingroup,' n = 7 and 'brief outgroup,' n = 8). To extract neural activity due to social interaction, non-trapped rats were exposed to a free rat across a wire mesh over three daily sessions ('2 free ingroup,' n = 8 and '2 free outgroup,' n = 7). In this control condition, both animals could freely explore the conspecific in a context that mimicked the level of contact with a trapped rat across the holes of the restrainer. Additionally, the neural activity of rats trapped inside the restrainer ('trapped,' n = 8) and non-tested rats ('baseline,' n = 10) was examined.

## A common neural response to the HBT independent of group identity

To identify patterns of neural activity associated with each condition in a minimally biased way, multivariate task partial least square analysis (PLS, see Materials and methods) was conducted on the six social conditions (*Figure 2D–F*). This analysis found a neural pattern that distinguished the HBT ingroup and outgroup conditions from the other conditions. A significant latent variable (LV) emerged (LV1, p<0.001), which was characterized by a contrast between brain-wide c-Fos expression in the HBT conditions and all other social control conditions (*Figure 2E*, left). This finding reflects increased overall neural activity in both HBT conditions compared to the control conditions (ANOVA $F_{(5,40)}$ = 21.04, p<0.001, Bonferroni, p<0.05, *Supplementary file 2*, *Figure 2—figure supplement 3*). Permutation and bootstrapping tests were used to identify brain regions that maximally contributed to this contrast between conditions. The relative contribution of each brain region (termed here 'salience,' *Figure 2E*, right), provides a profile of neural activation across tasks. Activity in multiple brain regions emerged as significantly salient for the HBT conditions (regions crossing the threshold line denote a p<0.01, *Figure 2E*, right). The frontal cortex and anterior insula (AI) received particularly high salience scores, indicating that high levels of neural activity in these regions were driving the contrast between the HBT ingroup and outgroup conditions, and the controls.

## Dissociating neural activity from environmental and motor confounds

Neural activity measured on the final testing session reflects both exposure to a trapped conspecific as well as an intention or lack thereof to open the restrainer, indicated by each rat's history of door-

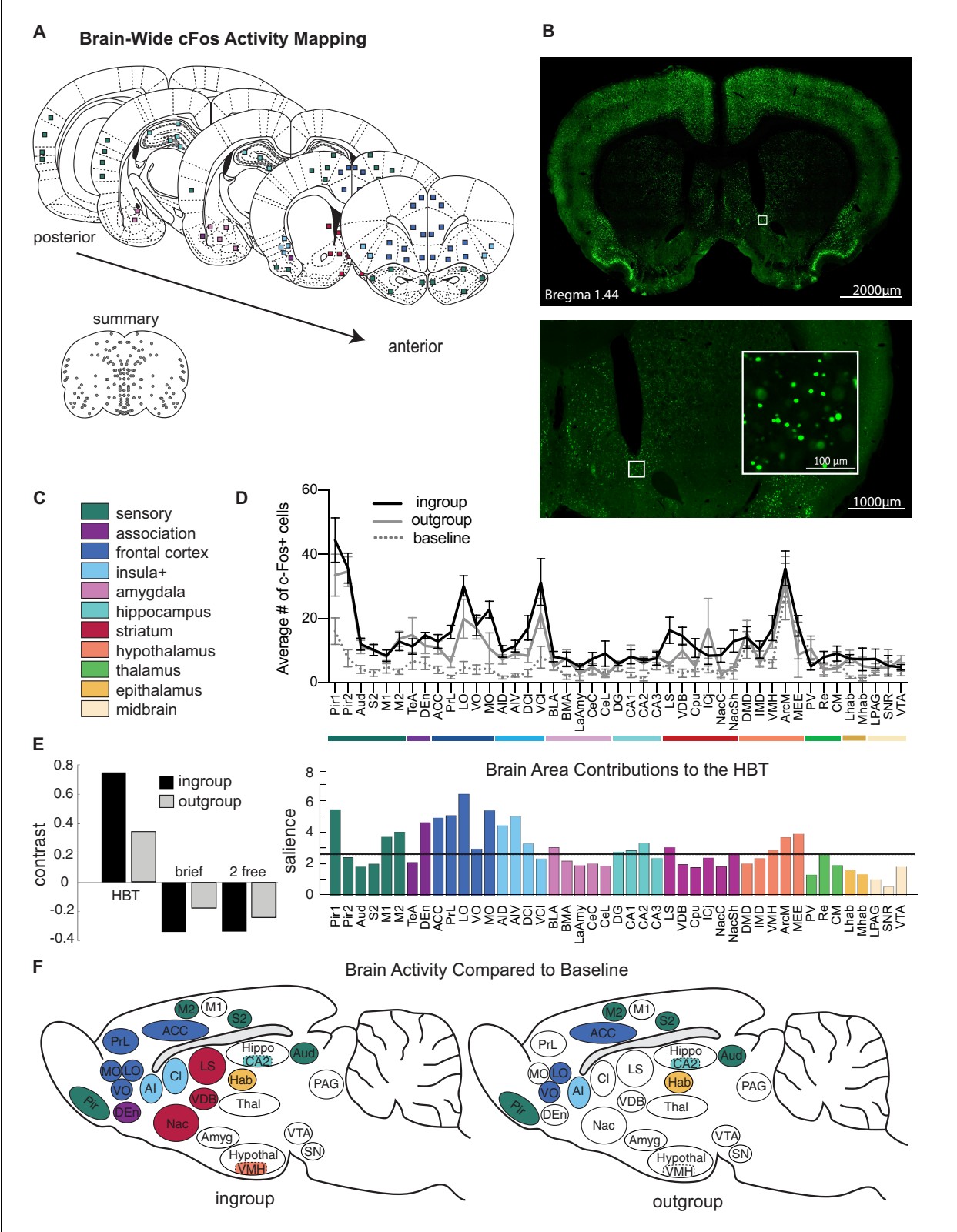

**Figure 2.** Neural activity associated with the helping test. Brain-wide patterns of neural activity associated with the helping behavior test (HBT). (**A**) Diagram of brain regions sampled for c-Fos expression. (**B**) A representative image of c-Fos signal sampled in the nucleus accumbens. (**C**) Legend of brain region categories coded by color. (**D**) Number of c-Fos⁺ cells per region (mean ± SEM) for rats tested with ingroup members, outgroup members, and an untested baseline. (**E**) Partial least square (PLS) task analysis of all social conditions. On left, the HBT ingroup and outgroup conditions showed a

*Figure 2 continued on next page*

*Figure 2 continued*

common pattern of neural activity, which contrasts with the other social conditions, including a brief exposure to a trapped rat (brief), and exposure to a non-trapped rat separated by a wire mesh (2 free). On right, regions that contributed to this contrast display increased activity in the HBT compared to the other conditions. The black line marks a significance threshold at p<0.01. (F) Diagram of rat brains showing regions significantly active (in color) for the HBT ingroup (left) or outgroup (right) conditions compared to baseline.

The online version of this article includes the following figure supplement(s) for figure 2:

**Figure supplement 1.** c-Fos acquisition.
**Figure supplement 2.** Outline of control conditions.
**Figure supplement 3.** Box plots of c-Fos data in all brain regions across all test groups.
**Figure supplement 4.** Comparison of the helping behavior test (HBT) to the brief conditions.
**Figure supplement 5.** c-Fos-associated movement patterns on the final testing session.
**Figure supplement 6.** Rats tested in the helping behavior test (HBT) with chocolate chips display little door-opening.

opening. To isolate the neural response associated with the HBT from neural activity associated with exposure to a trapped rat, each HBT condition (ingroup and outgroup) was compared with the corresponding brief condition (*Figure 2—figure supplement 4*). For animals tested with ingroup members, this analysis indicated increased brain-wide neural activity in the HBT compared to the brief condition, with numerous regions significantly contributing towards the contrast (*Figure 2—figure supplement 4A*). Yet, for animals tested with outgroup members, only a few regions significantly differed between the HBT and brief conditions, such as areas in the sensory cortices (*Figure 2—figure supplement 4B*). Furthermore, there was higher mean c-Fos for the ingroup relative to the outgroup in the HBT but not the brief condition (Bonferroni, p=0.02, *Figure 2—figure supplement 4C*). Thus, for rats tested with ingroup members, experiencing the HBT was more salient than brief exposure to a trapped rat. However, for rats tested with outgroup members, these two conditions were not dramatically different. As velocity and time spent around the restrainer were similar in the HBT ingroup and outgroup conditions on the final session (MMA, p>0.05, Bonferroni, p>0.05, *Figure 2—figure supplements 4D, E* and *5A–D*), the increased neural activity in the HBT ingroup condition is not likely due to motor movements. While c-Fos was not correlated with velocity on the final session (*Figure 2—figure supplement 5E*), it was found to be significantly correlated with time spent in the area near the restrainer (Pearson's r, p<0.05, *Figure 2—figure supplement 5F*). This finding provides further indication that motivational state, rather than increased motor activity, better explains the observed brain-wide neural activity.

The distinct activity in the HBT ingroup condition may reflect the rewarding nature of opening the restrainer. To distinguish this from the neural activity associated with a non-social reward, a separate group of rats was tested in the HBT with a restrainer containing five chocolate chips (*Figure 2—figure supplement 6A–C*). Unexpectedly, most rats did not learn to open the restrainer (two of eight became openers, *Figure 2—figure supplement 6D, E*), and rats tested with chocolate displayed significantly lower velocity (Bonferroni, p<0.001) and time around the restrainer (Bonferroni, p<0.01) across testing sessions compared to the HBT ingroup and outgroup conditions (*Figure 2—figure supplement 6F, G*). Moreover, brain-wide neural activity was significantly lower for this group than for the HBT ingroup and outgroup conditions (Bonferroni, p=0.01, p=0.03, respectively, *Supplementary file 2*, *Figure 2—figure supplement 3*). As door-opening was scarce in this condition, it is not informative about the neural activity associated with a non-social reward. However, it provides evidence that the neural activity observed in the HBT conditions was a response to the social context, rather than simply the experience of undergoing the full-length paradigm with a restrainer.

## A distinct neural signature of prosocial intent

The analysis above provided a broad overview of all social conditions, but was not informative about the neural activity associated specifically with either the HBT ingroup or outgroup conditions. To identify regions uniquely active in each HBT condition, c-Fos expression in the HBT ingroup and outgroup conditions was compared to the baseline of non-tested rats for each brain region. This

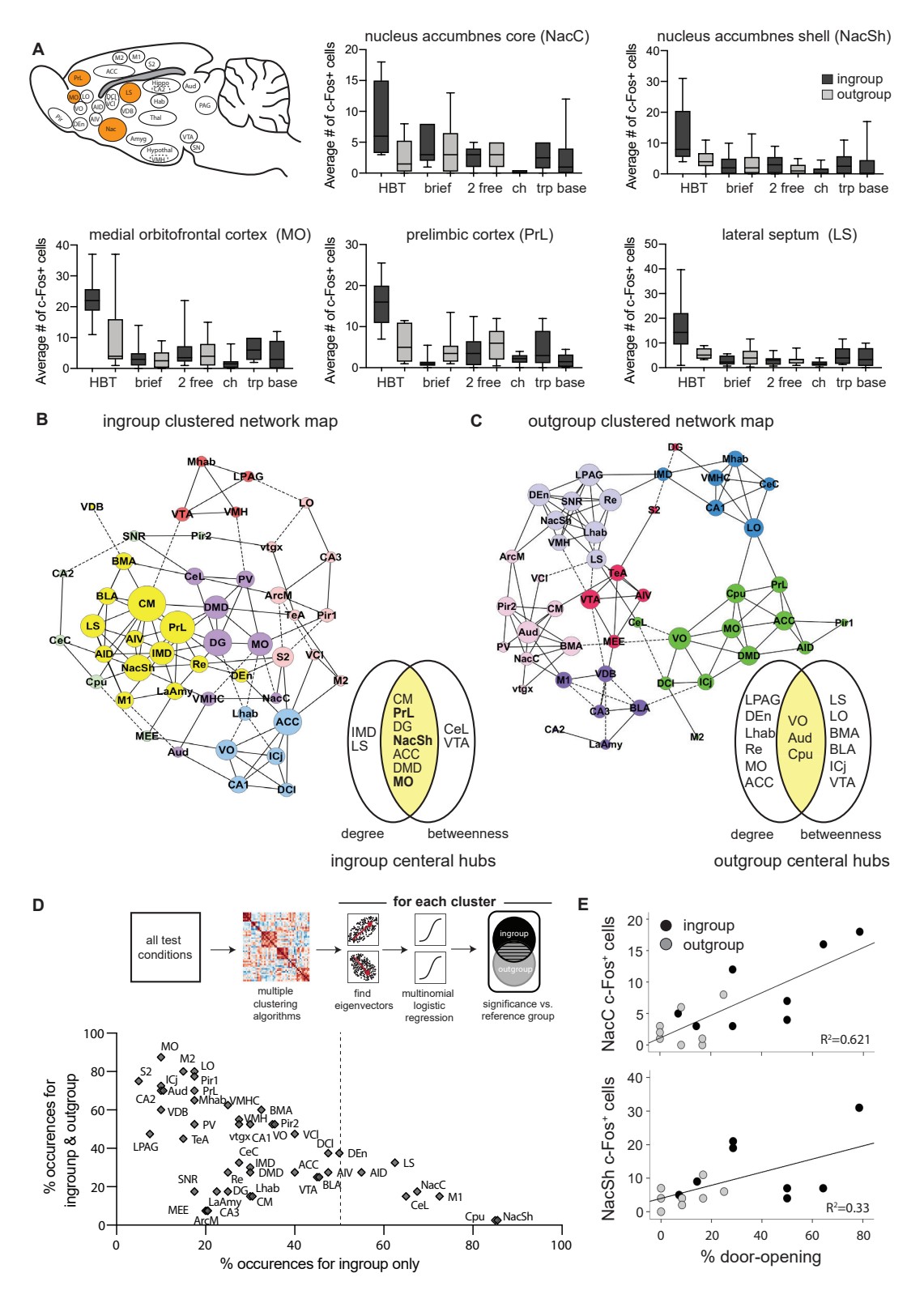

**Figure 3.** The nucleus accumbens (Nac) is selectively active for the helping behavior test (HBT) ingroup condition. The Nac was activated selectively for trapped ingroup members. (**A**) Several brain regions (in orange) were significantly more active in the HBT ingroup compared to the HBT outgroup condition (p<0.05). c-Fos numbers are also shown for the brief, 2 free, chocolate (ch), trapped (trp), and baseline (base) conditions. (**B, C**) Network graph depicting the top 10% inter-region correlations for the HBT ingroup and outgroup conditions. Positive correlations are shown in solid lines,

*Figure 3 continued on next page*

*Figure 3 continued*

negative correlations in dashed lines. Central hubs were determined as the top 20% of regions with highest in both degree and betweenness (yellow). In bold, regions that were more active in the HBT ingroup condition than the outgroup. Circle color represents clusters identified via a Louvain algorithm, circle size represents the number of degrees for each region. (D) A series of multiple logistical regression tests on all test conditions identified clusters of brain regions that aligned with the distinct brain activity in the helping test conditions. The figure contrasts regions uniquely observed for the ingroup condition (x-axis) with regions observed for both ingroup and outgroup conditions (y-axis). The nucleus accumbens shell (NacSh) and nucleus accumbens core (NacC) were present uniquely in the ingroup condition in 85 and 67.5% of tests, respectively. Dashed line represents the boundary for the regions that are required to identify the ingroup condition based on brain activity. Diagram describes how the graph was derived. (E) Activity in the NacC and NacSh was positively correlated with door-opening behavior. No other regions were significantly correlated with helping.

The online version of this article includes the following figure supplement(s) for figure 3:

**Figure supplement 1.** Network analyses.

---

analysis (*Figure 2F*) identified a common set of regions significantly more active than baseline (Bonferroni, $p<0.05$, *Supplementary file 3*) for both HBT ingroup and outgroup conditions, including the sensory cortex, AI, anterior cingulate cortex (ACC) and orbitofrontal cortex (ventral and lateral; VO and LO), CA2 of the hippocampus and the habenula (Hab). These regions are thus interpreted as participating in the response to a trapped rat, regardless of prosocial motivation.

A distinct set of regions was more active than baseline only in the HBT ingroup condition (Bonferroni, $p<0.05$, *Supplementary file 3*, *Figure 2F*). These included the medial orbitofrontal cortex (MO), prelimbic cortex (PrL), dorsal endopiriform cortex (DEn), nucleus accumbens (Nac), lateral septum (LS), claustrum (Cl), and the ventromedial hypothalamus (VMH). Of these areas, the MO, PrL, Nac, and LS were also significantly more active in the HBT ingroup compared to the HBT outgroup condition (Bonferroni, $p<0.05$, *Figure 3A*). These regions, which comprise part of the brains' reward and motivation network, showed low levels of activity in the control conditions (*Figure 3A*), indicating that the activity observed in the HBT was not due to social exposure or to the presence of a trapped rat. Rather, their specific activation in the HBT ingroup condition suggests that they play a role in the observed prosocial response to trapped ingroup members.

## Identifying central network hubs

To explore how different brain regions interact in the HBT ingroup and outgroup conditions, and to identify central hubs, functional connectivity was assessed based on c-Fos$^+$ cell counts. This strategy has previously proved useful for outlining the neural networks involved in complex behaviors like fear learning and other social behaviors (*Oliveira, 2013*; *Rogers-Carter et al., 2018*; *Vetere et al., 2017*; *Wheeler et al., 2013*). To this end, a covariance matrix based on c-Fos numbers was generated (*Figure 3—figure supplement 1A, B*) and clustered using a Louvain algorithm for community detection (*Blondel et al., 2008*). Network graphs thresholded at 10% of the top correlations were generated from these matrices to visualize the pairwise correlations, as well as provide input about the importance, or centrality, of each region to the network (*Sporns, 2013*; *Figure 3B, C*). This threshold was determined based on optimization of the network parameters such that it was as scale-free as possible without compromising connectivity and small-worldness (*Figure 3—figure supplement 1C–E*, see Materials and methods). Central hubs were identified as regions scoring in the top 20% for both degree (the number of connections) and betweenness (representing how many regions connect to others through this region; *Figure 3B, C*, *Figure 3—figure supplement 1F, G*). The nucleus accumbens shell (NacSh), PrL, and MO, areas identified above as selectively active in the HBT ingroup condition, also emerged as central hubs for the HBT ingroup network, suggestive of a functional role for these regions in the HBT ingroup condition.

As network graphs are prone to change based on selected parameters, a further analysis was conducted to enhance the validity of these findings. A series of 40 multinomial logistic regression tests compared the HBT ingroup and outgroup conditions to a reference group using varying parameters for threshold, clustering, and weighting of the network (see Materials and methods, *Figure 3D*, *Supplementary file 4*). With this alternative analysis, several regions emerged as

Photometry signal during the HBT

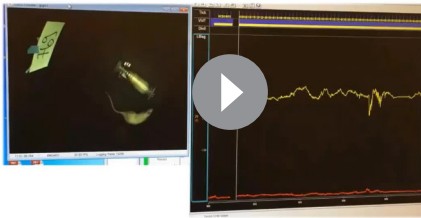

**Video 1.** Video demonstrating fiber photometry recording of a rat tested with an ingroup member. The video shows nucleus accumbens (Nac) activity increases when the free rat approaches a trapped ingroup member.
https://elifesciences.org/articles/65582#video1

significant only in the HBT ingroup condition. Specifically, the NacSh appeared in >80% of tests in clusters that were significant for the ingroup condition, but not the outgroup condition (*Figure 3D*).

In sum, neural activity in the Nac appeared to most distinctly characterize the HBT ingroup condition. Additionally, out of all brain regions tested, the Nac was the sole measured region where neural activity was significantly correlated with helping behavior, as expressed by the history of door-opening across testing days (*Figure 3E*). These findings led us to further explore the role of the Nac in vivo as rats experienced the HBT.

## Nac population activity in vivo demonstrates group selectivity

Across analyses, the Nac emerged as a central region for the HBT ingroup condition, both in activity levels and connectivity. In order to further explore Nac activity during the HBT, and to compare the Nac's response to a trapped ingroup and outgroup member within rats, calcium signal was recorded in vivo as an index of neural activity during the entire HBT for SD rats tested with trapped strangers of the same strain (ingroup, n = 8) via fiber photometry. Within-rat sessions included exposure to a trapped outgroup member (stranger of the LE strain), an empty restrainer, and an open arena as a baseline (see Materials and methods). As previously demonstrated (*Ben-Ami Bartal et al., 2014*), rats were motivated to help strangers of their own strain, and 5/8 of these rats became 'openers.' To tag firing neurons, an adeno-associated virus (AAV) driving the expression of the genetically coded calcium indicator, GCaMP6m, under the hSyn promoter was unilaterally injected into the right Nac, and an optic fiber was implanted at the same location (*Figure 4A, B*, *Figure 4—figure supplement 1*, see Materials and methods). Calcium signal was recorded by a photoreceptor and fluorescence intensity was analyzed as previously described (*Lerner et al., 2015*; *Figure 4C*). To measure activity during prosocial approach, instances of the free rat's entry into the area around the restrainer were identified as events of interest (*Figure 4D*). Nac activity significantly increased when rats approached a restrainer containing an ingroup member (expressed as Δf/F, n = 83 sessions, Wilcoxon ranked-sum test, p<0.05, *Figure 4E*, *Video 1*). In contrast, activity was not changed when these same rats approached a trapped outgroup member (n = 5 sessions), an empty restrainer (n = 47 sessions, Wilcoxon, p>0.05, *Figure 4E*), or when the rat was free to roam in an empty arena ('baseline,' n = 45 sessions, *Figure 4F*). As this measure is defined by a specific movement (entry into the zone around the restrainer), the motor movements associated with all events should be similar and are not a likely cause for the different neural signals across conditions. In evidence of this idea, velocity at the moment of entry into the restrainer was similar for the outgroup session and the following ingroup session (first 10 min, n = 7 per group, *Figure 4G*), and no differences were found in velocity (5.39 ± 0.28; 5.42 ± 0.34 cm/s), number of entries (19.1 ± 1.6; 19.6 ± 3.1), or time spent in the area around the restrainer (198.9 ± 18.7; 223.7 ± 29.1 s) between these ingroup and outgroup sessions, respectively (paired t-tests, p>0.05, mean ± SEM).

The above event-related analysis focused on the signal in the few seconds around the act of approaching the restrainer. Next, activity was analyzed over the duration of the entire session. For ingroup members, Nac activity was significantly higher when the free rat was located in the area around the restrainer compared to when they were outside this zone (repeated-measures ANOVA, p<0.01, Bonferroni p<0.01, *Figure 4H*). This effect was not observed for outgroup members. As the outgroup data represents activity recorded over a single session, the sampling is lower for this condition. Yet, each session included multiple approaches to the restrainer (total n = 248 entries into the zone), rendering this statistical analysis feasible.

To understand if the observed increase in signal was related to door-opening or to the intention to open the restrainer, activity at the moment of door-opening was analyzed. A peak in activity was

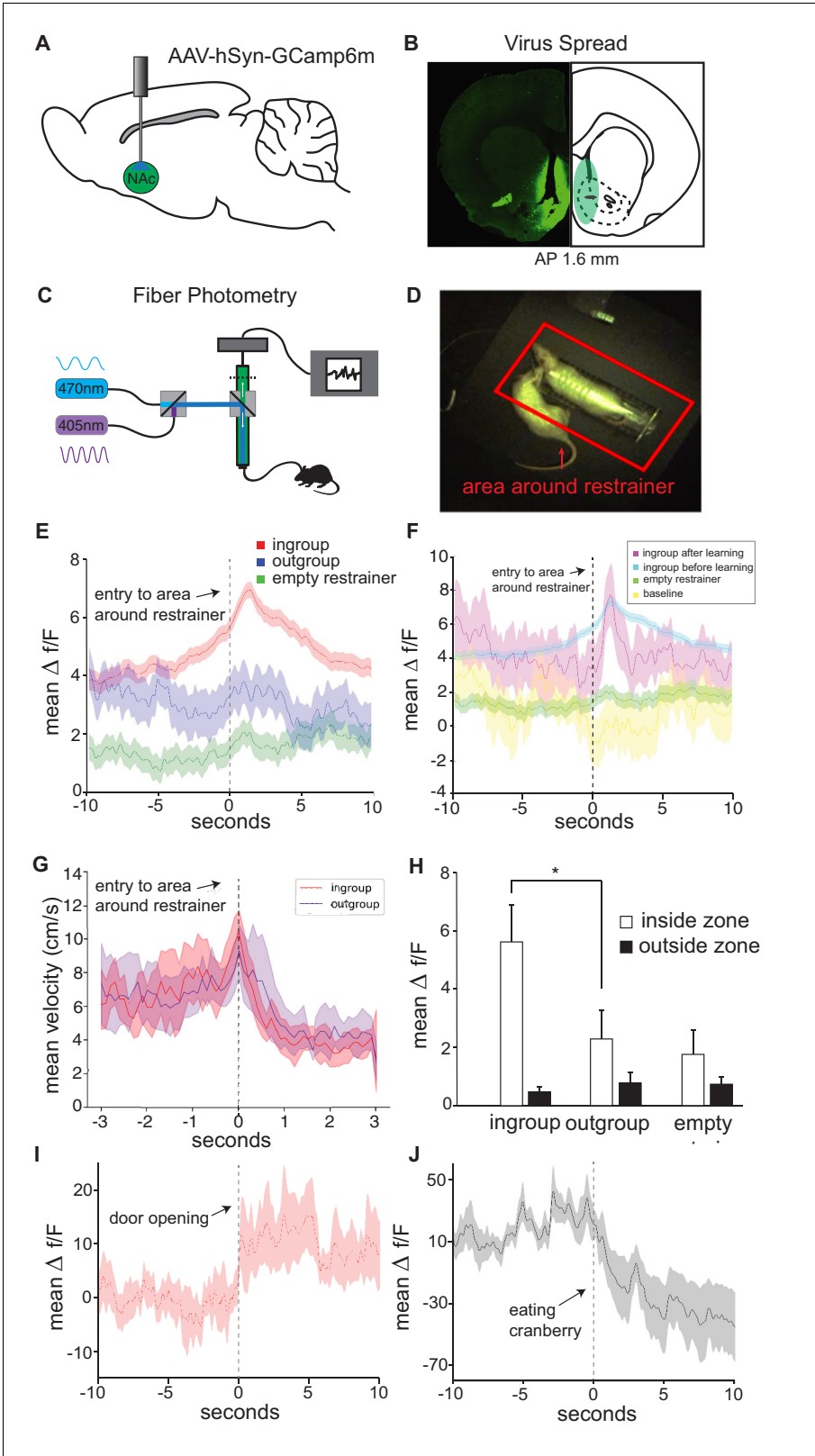

**Figure 4.** In vivo neural activity in the nucleus accumbens (Nac) corresponds with approach towards an ingroup member. Approaching a trapped ingroup member was associated with increased calcium signal in the Nac. (**A**) A diagram depicting location of virus injection and optic-fiber implant used for fiber photometry recordings. (**B**) Example of virus spread in one animal (left) and overlay summary of all animals (right). (**C**) Diagram of setup used
*Figure 4 continued on next page*

*Figure 4 continued*

for photometry recordings. (D) Top view of testing arena. The area around the restrainer is depicted by the red rectangle. Entry into the area, defined as when the rat's body was detected moving from outside this zone to inside this zone, was the point used in the analyses below. (E) Mean neural activity (Δf/F) across rats and testing sessions increased when rats approached a trapped ingroup member (red) but not a trapped outgroup member (blue) or empty restrainer (green). (F) Activity increased both during sessions before (cyan) and after (magenta) rats learned to open the restrainer. No increase was observed while rats were recorded in an empty arena (yellow). (G) Velocity during entry into the zone was not different across conditions (mean ± STDEV). Point of entry into the area around the restrainer is indicated by the dashed line. (H) Neural activity averaged across the whole session (mean ± SEM) was higher for the ingroup when the rat was in the area around the restrainer compared to outside this zone. Activity is also shown around the moment of door-opening (I) or when rats started eating a cranberry on the final session (J).

The online version of this article includes the following figure supplement(s) for figure 4:

**Figure supplement 1.** Summary of fiber photometry injections and implants.

observed at the moment of door-opening, indicating that door-opening itself was a salient event (*Figure 4I*). Yet, increased activity during approach was observed even before rats learned to open the restrainer (*Figure 4F*), indicating that Nac activity was implicated more generally in approach rather than in the act of door-opening itself. Lastly, to examine the Nac's neural response to a non-social reward, cranberries were placed in the restrainer on the last session. We found that activity significantly decreased when rats ate a cranberry, evidence that the Nac was active during seeking, rather than reward acquisition (n = 15 eating events, Wilcoxon, p<0.05, *Figure 4J*). These data provide further support for selective Nac activation for ingroup members and suggest that prosocial approach is associated with increased Nac activity.

## A subpopulation of cells projecting from the ACC to the Nac participates in prosocial approach

The data described above identified the frontal cortex as participating in the HBT, with the PrL and MO more active for ingroup members. To identify inputs from the frontal cortex to the Nac that promote prosocial behavior, neurons were co-labeled for c-Fos and a retrograde tracer. Structural projections were marked by injecting the retrograde tracer Fluoro-Gold (FG) into the Nac prior to participation in the HBT with a trapped ingroup member (n = 13, *Figure 5A, B*). Most animals learned to open the restrainer (8/13 became 'openers'). Co-labeling of FG$^+$ and c-Fos$^+$ cells identified cells that were active during the HBT and structurally projected to the Nac (*Figure 5C, D*). The frontal cortex, insula and BLA, all areas with known projections to the Nac, were sampled for co-labeling. The PrL and MO in particular showed substantial FG$^+$ labeling (*Figure 5E*). The frontal cortex also showed significantly more co-labeling for c-Fos and FG than the insula and BLA (ANOVA, $F_{(2,34)}$ = 20.1, p<0.001, *Figure 5F*), indicating that these inputs to the Nac were more active during the HBT. A significant positive correlation (Pearson's $r^2$ = 0.37, p=0.03) between the co-labeled cells and door-opening was uniquely observed in the projection from the ACC to the Nac (*Figure 5G*), whereas ACC c-Fos$^+$ cell numbers as a whole did not correlate with door-opening (*Figure 5H*).

A comparison between the openers (n = 8) and non-openers (n = 5) revealed overall increased c-Fos levels across sampled regions (ANOVA, $F_{(1,\ 88)}$ = 27.8, p<0.001, *Figure 6A*), with significantly higher c-Fos levels in the LO and MO of openers (Sidak correction, p<0.01), suggesting that the neural activity in the OFC may vary according to this behavior. Analysis of FG$^+$ cells also showed overall increased numbers for the openers (ANOVA, $F_{(7,88)}$ = 7.9, p<0.001, *Figure 6B*), yet no significant differences emerged per region. We noted the BLA as an interesting region for further exploration, with a markedly increased activity in non-openers (Sidak correction, p=0.07). This is suggestive of an increased structural connectivity between the BLA and the Nac in rats that did not show helping behavior towards ingroup members. Finally, no significant difference was identified in the percent of co-labeled FG$^+$/c-Fos$^+$ cells (ANOVA, p>0.05, *Figure 6C*). In conclusion, these findings, while

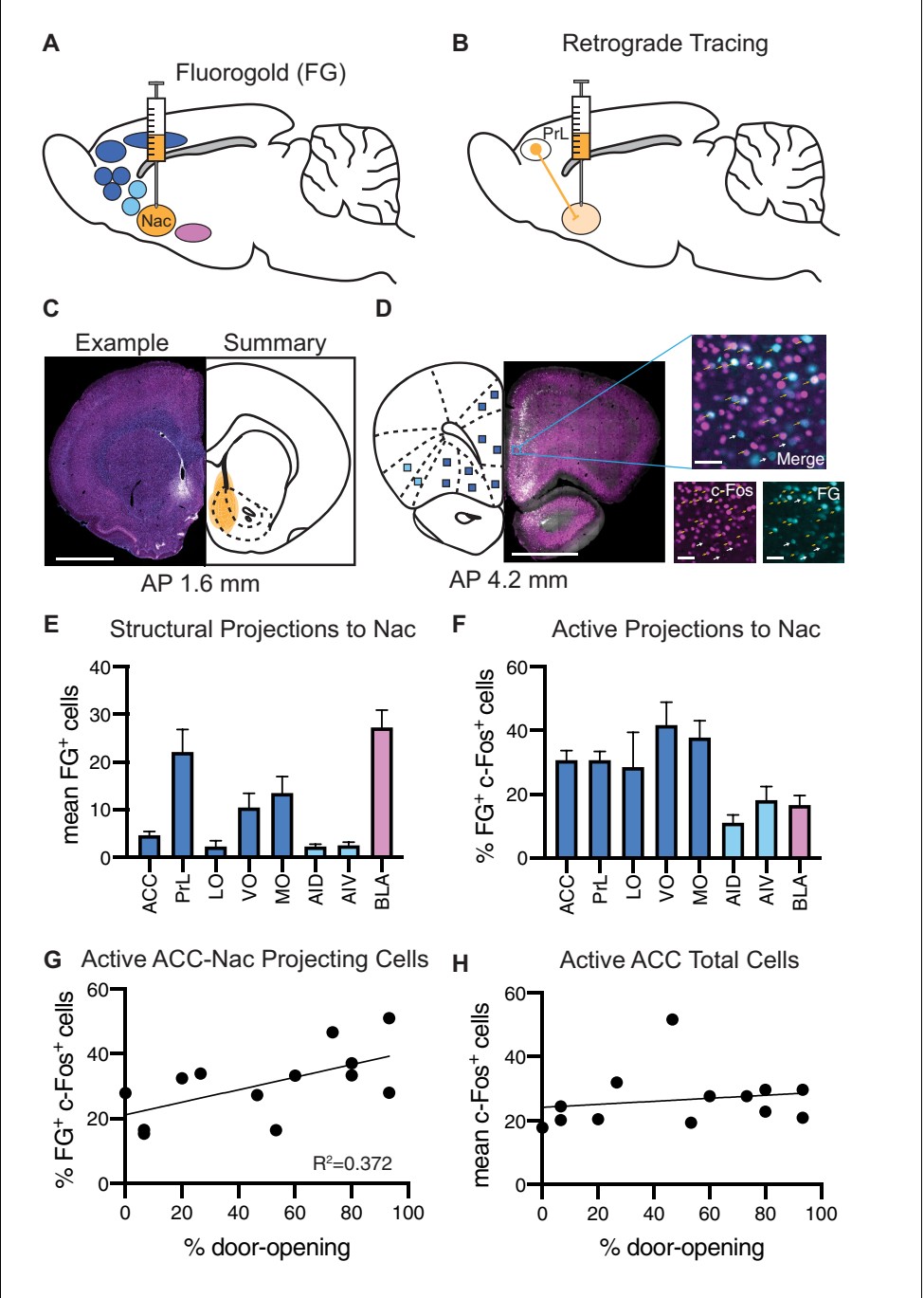

**Figure 5.** Analysis of structural projections to the nucleus accumbens (Nac) co-labeled for c-Fos. Structural inputs to the Nac and their corresponding activity during the helping behavior test (HBT). (**A**) Schematic of a retrograde injection into the Nac and the input regions that were analyzed. (**B**) Diagram of retrograde tracing labeling cells in the prelimbic cortex (PrL) that project to the Nac. (**C**) On left: example Fluoro-Gold (FG) injection. FG is in blue, c-Fos in magenta. On right: summary of all FG injections. Scale bar: 2 mm. Coordinates are anterior-posterior from Bregma *Paxinos and Watson, 1998*. (**D**) On left: an example coronal section containing regions of interest (ROIs) that to project to the Nac. On right: a fluorescent image of a PrL ROI containing FG+ cells (blue) and c-Fos+ cells (magenta). ROI scale bar: 50 µm. (**E**) Average number of FG+ cells for each brain ROI. (**F**) Percent of FG+ cells that were co-localized with c-Fos, for each region. (**G**) The percent of FG+ cells co-localized with c-Fos in the anterior cingulate cortex (ACC) positively correlated with the percent of door-openings across testing sessions. (**H**) The average number of c-Fos+ cells in the ACC did not correlate with the percent of door-openings.

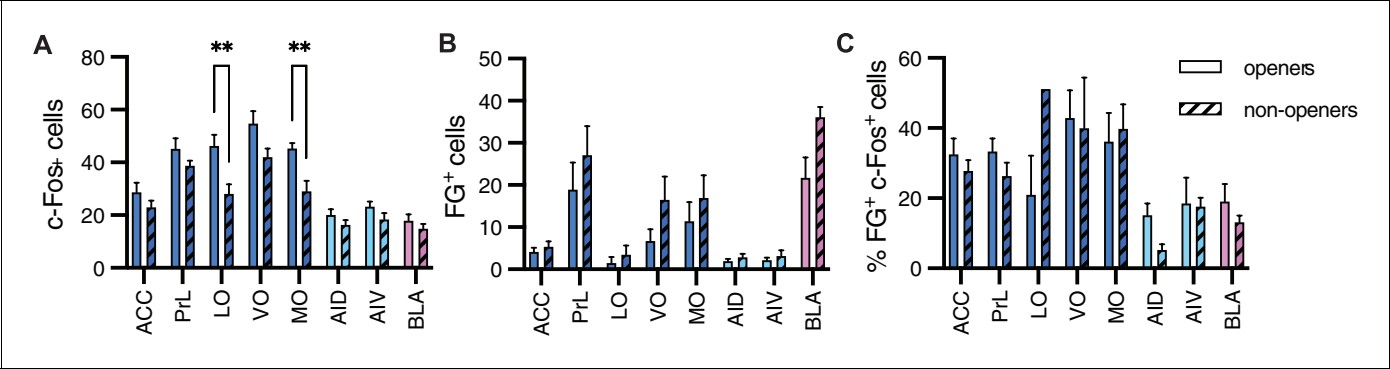

**Figure 6.** Comparison between openers and non-openers. Higher c-Fos levels in the orbitofrontal cortex were observed for openers than for non-openers. (A) A comparison of total c-Fos numbers between five non-openers and eight openers in the retrograde experiment for rats tested with ingroup members. No difference was significant for the total FG+ numbers signifying structural connections (B) or the percent of co-labeled c-Fos+/FG+ cells (C) for these populations.

descriptive and based on a small sample size, point to the ACC-Nac projection as a target for future investigation into the circuitry involved in prosocial behavior.

## Discussion

In these experiments, rats were tested for helping behavior with trapped rats from their ingroup, who typically elicit prosocial motivation, or with outgroup members, who do not. Increased neural activity was observed across multiple brain regions for rats in the HBT conditions compared to the control conditions. Regardless of the identity of the trapped rat, neural activity in the sensory, orbitofrontal, anterior cingulate, and insular cortices was observed, evidence that these regions participate in processing the distress of others but are not predictive of prosocial behavior. The frontal regions MO and PrL, as well as striatal regions LS and Nac, were significantly more active in the ingroup than the outgroup HBT conditions. The Nac, a region associated with reward seeking and social reward in rodents (*Aragona et al., 2006*; *Báez-Mendoza and Schultz, 2013*; *Dölen et al., 2013*; *Gunaydin et al., 2014*), emerged as a central hub of the ingroup network and correlated with prosocial approach and helping behavior, pointing to its recruitment in a prosocial context.

The activation of the Nac observed in this study mirrors a broad literature of the Nac's role in social reward-related behaviors. The Nac is mainly composed of GABAergic medium spiny neurons (MSNs) that fall into two classes: those expressing primarily dopamine receptor D1 (D1 MSNs) and those expressing primarily dopamine receptor D2 (D2 MSNs). Broadly, these pathways are thought to lead to opposing outcomes, with D1 MSNs and their pathways activated during reward, and D2 MSNs and their pathways activated during aversion (*Klawonn and Malenka, 2018*, but see *Soares-Cunha et al., 2016*). For example, Nac MSNs are associated with stress-induced anhedonia (*Bessa et al., 2013*), and mice displaying depressive-like behaviors following chronic stress have decreased excitatory input in D1 MSNs and increased excitatory input in D2 MSNs (*Francis et al., 2015*). Further, increased D1 MSN activity is predictive of animals that are resilient to stress-induced changes in social behavior (*Muir et al., 2018*). While the experiments presented here do not distinguish Nac cell types, given the dichotomy of D1 and D2 activity – that is, that D2 MSNs are typically involved in aversive behaviors, including avoidance (*Boschen et al., 2011*), and D1 MSNs are typically involved in rewarding behaviors including aggression seeking (*Aleyasin et al., 2018*; *Golden et al., 2019*) – we speculate that dopamine receptor D1 MSN cells are involved in prosocial intent. This hypothesis would require further testing and will be an interesting area for future study.

One interpretation of these findings is that the observed neural activity for ingroup members reflects 'empathic helping,' prosocial behavior motivated by empathic arousal in response to the other's distress. This interpretation is based on the prevalent view of empathy as a major driver of prosocial behavior (*Decety et al., 2016*; *Eisenberg and Miller, 1987*). Importantly, the term empathic arousal encompasses an emotional response to the other's distress as well as 'caring,' or concern for their well-being (*Batson, 2009*). Here, we observed common elements in the neural

response to trapped ingroup and outgroup members regardless of helping. This finding provides evidence that rats sense and respond to outgroup distress, even in the lack of prosocial intent.

An alternative explanation of door-opening behavior is that rats are acting primarily for social contact. While accumulating evidence shows that social contact is not required for helping in rats (*Ben-Ami Bartal et al., 2011*; *Cox and Reichel, 2020*; *Sato et al., 2015*), this interpretation cannot be ruled out by the present dataset. Undoubtedly, mammalian behavior is motivated by social reward, and rats experience social reward during this paradigm. However, the helping test is a stressful situation: restraint is a well-validated chronic stressor (*Glavin et al., 1994*; *Ottenweller et al., 1994*; *Paré and Glavin, 1986*; *Servatius et al., 2000*), and testing induces a stress response in both the free and the trapped rat, including secretion of the stress hormone corticosterone, defecation, and freezing (*Ben-Ami Bartal et al., 2011*; *Ben-Ami Bartal et al., 2016*). It is thus important to consider that the behavior of the free rat is influenced by the stress of the trapped rat. Even if social reward was not the primary motivation, it is likely that rats found social contact and door-opening itself rewarding, resulting in a reward anticipation signal. Another possibility is that since restrainers are latched on the final test day, neural activity of successful helpers reflects frustration at failure to open the door. Based on the data presented here, it is impossible to disentangle these motivations and the accompanying neural elements of reward anticipation and prediction error from the prosocial response. Yet the very presence of such a signal would be inherently prosocial, reflecting an emotional response selectively elicited for ingroup members: a positive response to social contact, and an aversive response to the inability to release the trapped rat. Whatever the motivational components may include, they are all part of the response underlying ingroup bias.

An alternative explanation of the increased neural activity observed in the ingroup condition is that it may reflect primarily motor activity associated with door-opening, stemming from increased movement in these conditions. Yet, it is important to keep in mind that c-Fos expression reflects neural activity only on the final testing session, when no door-opening occurred and the restrainers were latched. Notably, no differences in velocity or time around the restrainer were observed on the last session between the HBT ingroup and outgroup conditions. The measure of entry into the area around the restrainer used in the fiber photometry experiment by definition controls for movement as it depicts the same movement across conditions as the event of interest. Thus, while motor-related processing is part of the overall neural activity displayed by these rats, it is unlikely to be the primary explanation for the differences observed between groups.

Methodological limitations should also be considered. c-Fos is an indirect index of neural activity that does not provide direct access to neural firing and is suspected to be influenced by other neural events, such as plasticity (*Kovács, 2008*; *Minatohara et al., 2015*). Moreover, this index provides high spatial but low temporal resolution, which may be critical in a complex social situation where different events occur over an hour-long period. Finally, although c-Fos was sampled from 84 regions covering most major brain areas, it is still but a fraction of the entire brain. The results presented above are correlational in nature and descriptive only. Correlation data in and of itself is not sufficient evidence, especially given the limited sample size and possibility of false discoveries. Thus, these data provide a partial picture of the rats' neural response and should be interpreted with this caveat in mind. It is nonetheless encouraging that the Nac emerged as a key region across several measures. The increase in c-Fos[+] cells observed in the ingroup condition was mirrored by the in vivo calcium signal, and both measures were associated with behavior (helping and approach respectively). It is important to note that as the data presented here represents activity in male rats only, conclusions are limited to males. The collection of data from female rats is currently ongoing and will be important for generalizing these results. Overall, this article aims to provide a broad overview of the neural circuitry associated with prosocial intent and provides a base for future work that will aim to dissect these circuits and understand their causal contribution to behavior.

Regarding the motivational state underlying door-opening, consider the following: when rats open the restrainer quickly on consecutive days and become 'openers,' it is clearly an intentional behavior. However, when rats fail to open the restrainer, it could be explained by either lack of will or lack of ability to open the door. Here, some rats tested with ingroup members did not become 'openers.' Previous experiments similarly found that while most rats help ingroup members, around 30% of SD rats do not become openers. The current dataset does not provide enough information to support any conclusions regarding the motivational state of the 'non-openers' to surmise whether they were unable or unwilling to open the restrainer. Analysis presented above suggests that the

OFC and BLA are potential targets for future investigation of these two behavioral phenotypes. Individual variability of helping behavior is of great interest and needs to be investigated in depth in future studies.

Several cortical regions exhibited graded activity in the HBT. While they were most active in the ingroup condition, they were also notably more active for HBT outgroup members compared to the baseline conditions. This may indicate that a threshold of activation must be crossed to initiate prosocial approach. Alternatively, it may point to specific neural populations within these regions that are involved in helping. In the ACC, total c-Fos levels were not correlated with helping, yet the subset of c-Fos$^+$ cells projecting to the Nac was positively correlated with helping. This suggests that at least for the ACC the increased activity observed in the ingroup condition is due to the recruitment of a specific neural population. Emerging technologies enabling cell-specific recordings are opening a path for future studies to further identify circuit and cell-type-specific activation involved in the behavior we observed. These technologies, along with further investigation, will allow researchers to expand upon the current findings to a point that may permit artificial manipulation of prosocial motivation.

It is a major goal for society to understand the empathy gap for outgroup members: why do we help some, but remain impervious to the suffering of others? The rats we tested demonstrated social selectivity; they did not help a rat of an unfamiliar strain. The study of rodents allows us to examine the neural activity that leads to this behavior. A growing body of work demonstrates that empathy-like behavior in rodents, including vicarious fear and contextual fear learning, is associated with neural activity in the ACC and insula (*Carrillo et al., 2019*; *Hernandez-Lallement et al., 2020*; *Jeon et al., 2010*; *Karakilic et al., 2018*; *Lin et al., 2018*; *Raam et al., 2017*; *Rogers-Carter et al., 2018*; *Sakaguchi et al., 2018*; *Sato et al., 2015*; *Schaich Borg et al., 2017*; *Zheng et al., 2020*). The similarity of these findings with the human research is notable, where empathy to others' pain has been linked with ACC and insular activity (*Preston and de Waal, 2002*; *Shamay-Tsoory and Lamm, 2018*). The findings presented here, showing Nac activation during a prosocial context, are congruent with demonstration of Nac activity during helping and compassion in humans (*Hackel et al., 2017*; *Harbaugh et al., 2007*; *Inagaki et al., 2016*; *Klimecki et al., 2014*; *Lamm et al., 2011*; *Morelli et al., 2018*). In conclusion, this study provides the first evidence for a common neurobiological mechanism driving empathic helping across mammalian species and highlights a distinct neural response to the distress of affiliated others. These findings provide insight into the way the brain determines the value of others' outcomes based on their social identity and open a path towards predicting and influencing prosocial decisions.

# Materials and methods

## Key resources table

| Reagent type (species) or resource | Designation | Source or reference | Identifiers | Additional information |
|---|---|---|---|---|
| Strain, strain background (*Rattus norvegicus*) | Sprague–Dawley Rat | Charles River Labs | Charles River 001; RRID:RGD_10395233 | |
| Strain, strain background (*Rattus norvegicus*) | Long–Evans Rat | Envigo | Envigo: HsdBlue:LE; RRID:RGD_5508398 | |
| Recombinant DNA reagent (virus strains) | AAV-hSyn-GCaMP6m | Addgene | Addgene: 131004; RRID:Addgene_131004 | |
| Antibody | Rabbit anti-cFos primary antibody | Millipore Sigma | Millipore: ABE457; RRID:AB_2631318 | IHC (1:1000) |
| Antibody | Donkey anti-rabbit IgG Alexa Fluor 488 secondary antibody | Jackson Immuno Research Labs | Cat#: 711-545-152; RRID:AB_2313584 | IHC (1:500) |
| Chemical compound, drug | Fluoro-Gold | Fluorochrome | RRID:AB_2314408 | |

*Continued on next page*

*Continued*

| Reagent type (species) or resource | Designation | Source or reference | Identifiers | Additional information |
|---|---|---|---|---|
| Software, algorithm | MATLAB | MathWorks | RRID:SCR_001622 | |
| Software, algorithm | Zen | Zeiss | RRID:SCR_013672 | |
| Software, algorithm | Fiji | NIH | RRID:SCR_002285 | |

## Experimental design

### Animals

Rat studies were performed in accordance with protocols approved by the Institutional Animal Care and Use Committee at the University of California, Berkeley. Rats were kept on a 12-hr light-dark cycle and received food and water ad libitum. In total, 83 rats were tested across all experiments. Adult male SD rats ('SD,' age p60–p90 days) were used as the free and trapped ingroup rats (Charles River, Portage, MI). Adult male Long–Evans rats were used as trapped outgroup rats ('LE,' Envigo, CA). All rats that were ordered were allowed a minimum of 5 days to acclimate to the facility prior to beginning testing.

### Helping behavior test (HBT)

The HBT was performed as described previously (*Ben-Ami Bartal et al., 2011*). Animals were handled for 5 days and tested for boldness four times. For 3 days, animals received 30 min habituation sessions to the arena, followed by a 15 min session of open-field testing in the same arena on the fourth day. The boldness and open-field testing were conducted in order to remain consistent with previous experiments, but were not analyzed and are thus not reported.

During the HBT, a free rat was placed in an open arena containing a rat trapped inside a restrainer (*Figure 1A, D–F*). The free rat could help the trapped rat by opening the restrainer door with its snout, thereby releasing the trapped rat. One-hour-long sessions were repeated daily over a 2-week period. At 40 min, the door was opened half-way by the experimenter; this was typically followed by the exit of the trapped rat and was aimed at preventing learned helplessness. Door-opening was counted as such when performed by the free rat before the half-way opening point. Rats that learned to open the restrainer and consistently opened it on the final three days of testing were labeled as 'openers.'

To prevent individual familiarity with the outgroup member, strangers were swapped daily. Note that we previously found rats will release trapped strangers of their own strain, and that familiarity with the strain determines prosocial behavior rather than individual familiarity (*Ben-Ami Bartal et al., 2014*). On the last day of testing, the restrainer was latched shut throughout the 60 min session and rats were perfused immediately following behavioral testing. Sessions were video recorded with a CCD color camera (KT&C Co, Seoul, Korea) connected to a video card (Geovision, Irvine, CA) that linked to a PC. Movement data were analyzed using Ethovision video tracking software (Noldus Information Technology, Inc, Leesburg, VA). Housing conditions: rats were housed in pairs, with a cagemate of the same strain, and were allowed 2 weeks to habituate to their cagemate prior to any testing. In the c-Fos experiment, ingroup members were SD cagemates. For the photometry experiments, ingroup members were SD strangers. The trapped outgroup member was a different LE stranger daily.

## Control conditions

For the 'chocolate' condition, rats underwent same procedure as the HBT described above, but instead of a rat, five chocolate chips were placed inside the restrainer. To prevent novelty-induced hypophagia, rats were exposed to chocolate chips in their homecage (and consistently ate them) a few days before start of testing. In the 'brief' condition, rats were tested for three daily 1-hr-long sessions with a rat trapped inside a latched, unopenable restrainer. In the '2 free' condition, a wire mesh was placed in the arena with the test rat on one side and the partner on the other side. Neither rat was trapped. For the 'trapped' condition, rats trapped inside the restrainer were used following the last session with ingroup members. For the 'baseline,' adult SD rats were used. These

rats did not undergo behavioral testing and, thus, c-Fos measurements reflect time in the homecage.

## HBT for the fiber photometry experiment

A variation on the HBT was performed to allow a within-subject comparison to the control conditions. Following a week of handling, rats were tested with a trapped rat of the same strain for daily sessions while connected to the optic fiber. The door was opened half-way by the experimenter after 30 min. Rats usually escaped the restrainer within a few minutes of half-way door-opening. After the trapped rat exited the restrainer, rats were allowed 10 minutes of free roaming, before the trapped rat's removal. Then, 10 minutes of recording were conducted while the free rat was in the arena with the empty, closed restrainer. Finally, the restrainer was removed and another 5 min baseline was recorded for the free rat in the empty arena. Additionally, rats were tested with a trapped outgroup member, an LE rat, on day 7. None of the rats had learned to open the restrainer at this point. We hypothesized that the optic cable was an obstacle to door-opening. Thus, the paradigm was run without the optic fiber for a week, allowing rats to learn the opening behavior. For this, rats were given two 30 min sessions of habituation time in the arena without the fiber, followed by 4–5 days with the trapped rat, as described above. As the fiber was not connected, no neural signal was collected during these sessions, which were recorded by video. Rats were then tested with the fiber for another 3 days with the trapped rat. We found that rats that had learned the door-opening behavior continued to do so on the later sessions when the wire was now connected.

On the last session, five rats were tested with a dried cranberry placed inside the closed restrainer in order to record the neural activity involving a non-social reward. Once rats ate the cranberry (within a few minutes), another cranberry was placed inside the restrainer by the investigator and the door was closed again. This was repeated three times per animal. For one rat that did not open the restrainer, the cranberries were placed in the arena outside the restrainer. Rats always ate all cranberries offered.

### Immunohistochemistry

On the last day of testing, animals were sacrificed within 90 min from the beginning of the session, at the peak expression of the early immediate gene product c-Fos. Rats were transcardially perfused with 0.9% saline and freshly made 4% paraformaldehyde in phosphate buffered saline (PBS). Brains were sunk in 30% sucrose as a cryoprotectant and frozen at –80°C. They were later sliced at 40 μm and stained for c-Fos. Sections were washed with 0.1 M tris-buffered saline (TBS), incubated in 3% normal donkey serum (NDS) in 0.3% TritonX-100 in TBS (TxTBS), then transferred to rabbit anti-c-Fos antiserum (ABE457; Millipore, 1:1000; 1% NDS; 0.3% TxTBS) overnight. Sections were then incubated in Alexa Fluor 488-conjugated donkey anti-rabbit antiserum (AF488; Jackson, 1:500; 1% NDS; 0.3% TxTBS). Sections were briefly washed in 0.1 M TBS again. Sections were further stained in DAPI (1:40,000) for 10 min if they did not contain the retrograde tracer FG and were then washed for a further 15 min (3 × 5′). Tissue that contained FG could not be stained for DAPI as both dyes are excited by UV fluorescence and their spectra overlap. Lastly, all slides were coverslipped with DABCO, dried overnight, and stored at 4°C until imaged.

Immunostained tissue was imaged at 10× using a wide-field fluorescence microscope (Zeiss Axi-oScan) and was processed in Zen software. Regions of interest (ROIs) (250 × 250 μm squares) were placed across the whole brain (*Figure 2—figure supplement 1*) and closely followed the methods performed in *Wheeler et al., 2013*; *Sadananda et al., 2008*. A custom written script in ImageJ V2.0.0 (National Institute of Health, Bethesda, MD) was used to quantify immunoreactive nuclei (either c-Fos$^+$ and/or FG$^+$ cells), followed by manual checks and counting by multiple individuals who were blind to condition; consistency for counts across individuals was verified by a subset of samples. The threshold for detection of positive nuclei was set at a consistent level for each brain region, and only targets within the size range of 25–125 mm$^2$ in area were counted as cells. Manual verification was targeted at identifying gross errors in the ImageJ scripts. For instance, in some cases the script falsely identified >100 cells within the counting square; this usually occurred when there was high background staining. This type of error occurred in ~15% of the samples, which were then manually corrected. All means are reported as mean ± SEM. Furthermore, 39 values for cell counts were removed from the dataset as outliers. The outliers were defined as those that were more than

2 standard deviations higher or lower than the group mean and further fell outside of the observed range for all conditions.

## Network analysis

### Generating network graphs

The network graph was obtained via a correlation matrix of c-Fos activity between all brain regions (*Figure 3—figure supplement 1A, B*). The top correlations are presented in a graphic form (*Figure 3B, C*). The cutoff threshold of 10% top correlations was determined based on scale-free network characteristics (*Figure 3—figure supplement 1C–E*). To obtain network graphs of the c-Fos data, pairwise Pearson's correlation coefficients were determined between the number of c-Fos$^+$ cells for all pairs of brain regions. The top 10% correlations from these matrices were used to generate network maps (*Figure 3B, C*). Correlation values higher than the cutoff were set to 1, and the corresponding brain regions >1 were considered connected for that threshold cutoff. This resulted in a brain region network for each correlation threshold cutoff.

### Calculating the correlation threshold

To obtain this threshold, all the possible correlation thresholds were enumerated and applied to the correlation matrices of the c-Fos data. Scale-free topology index and percent connectivity were computed for each network based on the WGCNA tutorial (*Langfelder and Horvath, 2008*). The results were plotted as functions of correlation threshold cutoffs. A positive value of the scale-free topology index indicates a scale-free network, a property of diverse types of non-random networks (*Barabási, 2009*). A correlation threshold cutoff of 10% was the transition point where the network demonstrated scale-free network characteristics (with a positive scale-free topology index), whereas a higher cutoff resulted in an unstable network with few connections. Additionally, the small-worldness of the brain networks was examined for the ingroup and outgroup maps. For this, the entire range of possible correlation threshold cutoffs was enumerated to compute two metrics: small-worldness and network density. The small-worldness was calculated in the same way as is discussed in *Humphries and Gurney, 2008*; the network density as is defined in *Wasserman and Faust, 1994*.

### Identification of central hubs

Central hubs were determined by ranking all brain regions according to two parameters: the betweenness value, representing the number of times all regions must pass through the ROI in order to reach other regions via the shortest path, and the number of connections (degrees) with other regions. The 20% top-ranking regions were then identified; brain areas that were in the top 20% for both categories were considered to be central hubs of the network (*Figure 3—figure supplement 1F, G*).

### Additional strategy for identifying central regions

The network presented is based on one set of parameters (threshold, clustering algorithm, weights), and as such, has limited validity. In order to increase the robustness of our conclusions, an additional analysis was conducted using an alternative strategy. The aim was to identify regions important for the ingroup condition according to which areas were unique actively in that region according to a pool of different statistical tests based on different parameters and configurations. To this end, a series of 40 multinomial logistic regression tests (*Figure 3D*) compared the HBT adult ingroup and outgroup conditions to a reference group (which was itself varied) using varying parameters for threshold, clustering, and weighting of the network for each test (*Supplementary file 4*). This method identified clusters of regions that repeatedly emerged as different than the baseline (reference group) clusters. Across all these tests, clusters varied slightly as the clustering algorithms are sensitive to the manipulated parameters. The number of times each brain region in the clusters significantly differed from baseline was quantified for each condition. To be more specific, in our customized ensemble clustering algorithm based on methods described in *Vega-Pons and Ruiz-Shulcloper, 2011*, each run of the 40 tests resulted in a computation of principal components. The first principal component (eigenvector) of each cluster was recorded. These were considered as the representative 'eigen region' for the clusters they were derived from. Subsequently, the eigen regions, representative of each cluster, were fed into a multinomial logistic regression

(*Hilbe, 2009*; *Zalaquett and Thiessen, 1991*) to determine whether they were significantly different between the ingroup and the reference group or the outgroup and the reference group. The reference group itself varied between using the untested condition and a broader reference group containing the c-Fos data from rats in the brief condition, rats in the '2 free' condition, the trapped rats themselves, and rats in the 'baseline' condition (total n = 48). The standard p-value cutoff of 0.05 was used after Bonferroni correction. Louvain clustering (*Blondel et al., 2008*) was done with the igraph package in R (*Csardi and Nepusz, 2006*), and Dynamic tree cut of hierarchical clustering was done with the WGCNA package in R (*Langfelder and Horvath, 2008*). 'Soft power' was defined as described in *Langfelder and Horvath, 2008*. Each cluster was classified as significantly different than the reference for the ingroup, outgroup, both, or none. The number of occurrences per region in each category was quantified. A high ratio of occurrences in the ingroup-only category indicates that this region was uniquely important for the ingroup condition.

## Fiber photometry calcium signal recordings

Rats underwent unilateral injections of 1 µL of virus (AAV-hsyn-GCaMP6m) into the right hemisphere of the Nac (AP: +2.0, ML: +1.0, DV: –7.2) and were implanted with an optic fiber patch cord with a numerical aperture of 0.48, and 400 µm core (Doric Lenses). Six weeks were allowed for virus infection, and upon signal detection, rats began testing in the HBT while neural activity was recorded, as described below. Rats were removed from the experiment in cases where no signal was detected or due to failed implants (n = 10). The protocol followed that previously performed in *Lerner et al., 2015*. An LED emitting 470 nm light was used for the $Ca^{2+}$ signal and 405 nm light was used as a control signal to remove movement artifacts. GCaMP fluorescence was collected by the same fiber; light passed through a dichroic lens with a GFP emission filter and was registered by a photoreceiver. Synapse software (TDT) and MATLAB code were used to demodulate the brightness from the 470 nm and 405 nm excitation and synchronize it with the video data. For analysis, signal was normalized to the median of the session and was represented as Δf/F after a least-square linear fit was applied via a custom MATLAB code. After testing was complete, rats were transcardially perfused with 0.9% saline and 4% paraformaldehyde in PBS. Brains were sunk in 30% sucrose as a cryoprotectant, frozen at –80°C, and sliced at 40 µm. Representative sections containing the Nac were mounted on slides and imaged in order to determine the location of the implant and virus spread. In order to line up entry to the point around the restrainer, the synchronized videos were analyzed in Ethovision (Noldus Information Technology, Inc). Using MATLAB code, each frame of entry (where 'in the zone' changed from 0 to 1) was identified, matched to the neural data via the synchronized time stamps, and used as the 0 point for *Figure 4E, F*, indicating entry into the zone around the restrainer. The time stamp of door-opening events was identified manually and used as input for the MATLAB code. Movement data was analyzed for the first 10 min of two subsequent sessions, the outgroup session and an ingroup session. This time frame was selected since for all sessions the restrainer was closed during this time.

## Retrograde tracing

Rats were anesthetized with 3–5% isoflurane and mounted onto a stereotaxic frame. The skull was exposed and a small hole was made above the determined stereotactic coordinates on the right hemisphere (AP: +2.0, ML: +1.0, DV: –7.2; from Bregma; *Paxinos and Watson, 1998*). A Hamilton Syringe containing the retrograde tracer Fluoro-Gold (FG, Fluorochrome, 4% in saline) was used to administer 200 nL of dye into the Nac. Rats were allowed 1–2 weeks to recover from surgery prior to starting the behavioral task. One animal failed to recover following surgery due to an infection and was removed from the experiment, bringing the total number of animals to 13. Following behavior, histology was performed as described above. The number of immunostained cells co-labeled for FG and c-Fos was manually counted in ImageJ (NIH), as well as the overall number of FG[+] cells. Co-labeled cells represent neurons that were active during the task and that project to the Nac. The number of c-Fos[+] cells was counted using an ImageJ script and then manually checked by hand. A subset of images was counted by two or more experimenters in order to ensure that cell counting was consistent across all observers.

## Statistical analysis

### Task PLS analysis

Task PLS is a multivariate statistical technique that was used to identify optimal patterns of neural activity that differentiate between the experimental conditions (*McIntosh, 1999*; *McIntosh et al., 1996*). It searches for LVs that can explain a large percent of the variability in the data by maximizing the contrast between test conditions. Through singular value decomposition, PLS produces a set of mutually orthogonal LV pairs. One element of the LV depicts the contrast, which reflects a commonality or difference between conditions. The other element of the LV, the relative contribution of each brain region (termed here 'salience'), identifies brain regions that show the activation profile across tasks, indicating which brain areas are maximally expressed in a particular LV. Statistical assessment of PLS was performed by using permutation testing for LVs and bootstrap estimation of standard error for the brain region saliences. For the LV, significance was assessed by permutation testing: resampling without replacement by shuffling the test condition. Following each resampling, the PLS was recalculated. This was done 500 times in order to determine whether the effects represented in a given LV were significantly different than random noise. For brain region salience, reliability was assessed using bootstrap estimation of standard error. Bootstrap tests were performed by resampling 500 times with replacement, while keeping the subjects assigned to their conditions. This reflects the reliability of the contribution of that brain region to the LV. Brain regions with a bootstrap ratio >2.57 (roughly corresponding to a confidence interval of 99%) were considered as reliably contributing to the pattern. Missing values were interpolated by the average for the test condition. An advantage to using this approach over univariate methods is that no corrections for multiple comparisons are necessary because the brain region saliences are calculated on all of the brain regions in a single mathematical step. MATLAB code for running the Task PLS analysis is available for download from the McIntosh lab website.

### Other statistical tests

In addition to the PLS analysis described above, a one-way ANOVA was conducted on the c-Fos data to compare the HBT ingroup and outgroup conditions and baseline for each brain region. Unpaired t-tests and two-way ANOVAs were used to compare the pattern of animals' movements. Bonferroni or Sidak post-hoc corrections were used following all ANOVAs. The comparison of velocity and time around the restrainer per day or minute was conducted via a MMA with an autoregressive covariance structure, with 'day' and 'minute' as the repeated measure for calculations of velocity and time around the restrainer. Movement was binned per minute. Data lines with velocity >30 cm/s were excluded as these values are not physically plausible and likely represent acquisition errors. To compare the photometry signal in the time before and after entry to the area around the restrainer, the signal in the second before entry to the zone (the 0 point) was compared to the signal in the second after entry to the zone around the restrainer for the ingroup, outgroup, and empty restrainer data. Wilcoxon signed-rank, a non-parametric test, was used in order to avoid assumptions about signal distribution. The comparisons of total session time spent in and outside the zone around the restrainer were conducted with a repeated-measures ANOVA, where the repeated measure was time in the zone and time outside the zone, as these were dependent samples for each animal. Note that the dataset used for the time-series analysis and the total time analysis are not the same, and thus the y-axis values are different. The time-series PSTH graph includes only the 10 s before and after each point of entry, whereas the total time bar graphs include the entire dataset for each session, split by location. Changes across days to helping behavior, including percent door-opening and latency to door-opening, were examined using the non-parametric Cochran's Q test and Friedman test respectively. Pearson's correlations were used for all correlation analyses; corrections for multiple comparisons were not run so as to avoid type 2 errors correcting for correlations with 45 brain regions.

### Number of animals

We aimed for eight animals per testing group. This number is based on our previous experience with the HBT. We generally observe ~70% of rats become 'openers' when tested with ingroup members (*Ben-Ami Bartal et al., 2011*) and expect no 'openers' for rats tested with outgroup members

(*Ben-Ami Bartal et al., 2014*); eight rats per group provide a power of 95% for incidence of 'openers,' with $\alpha$ and $\beta$ of 0.05.

## Number of brain regions sampled
The list of brain regions was based on previous work that followed the same methodology as used here (*Vetere et al., 2017*; *Wheeler et al., 2013*).

## Acknowledgements

We thank the following people for their help with this manuscript: Erin Aisenberg, Vanessa Alschuler, Noopur Amin, Anjile An, Kelsey Clausing, Dr. Robert Froemke, Pooya Ganjali, Nikita Gourishetty, Kelsey Hu, Kylie Huang, Ben Kantor, Claire Lee, Catriona Lewis, Dr. Chris Morrissey, Charu Ramakrishnan, Dr. Peter Sudmant, Yuanqui Sun, Charlie Walters, and Shannon Wong-Michalak. Imaging was conducted at the CRL Molecular Imaging Center, supported by the UC Berkeley Biological Faculty Research Fund. We would like to thank Holly Aaron and Feather Ives for their microscopy training and assistance. This manuscript was supported by the Miller Institute for Basic Science, Israel Science Foundation, Azrieli Foundation (IBB), and CIFAr (DKa).

## Additional information

### Funding

| Funder | Grant reference number | Author |
|---|---|---|
| Adolph C. and Mary Sprague Miller Institute for Basic Research in Science, University of California Berkeley | | Inbal Ben-Ami Bartal |
| CIFAR | | Daniela Kaufer |
| Israel Science Foundation | 0610719821 | Inbal Ben-Ami Bartal |
| Azrieli Foundation | | Inbal Ben-Ami Bartal |

The funders had no role in study design, data collection and interpretation, or the decision to submit the work for publication.

### Author contributions
Inbal Ben-Ami Bartal, Conceptualization, Data curation, Formal analysis, Funding acquisition, Investigation, Visualization, Methodology, Writing - original draft, Project administration, Writing - review and editing; Jocelyn M Breton, Formal analysis, Investigation, Visualization, Writing - review and editing; Huanjie Sheng, Formal analysis, Visualization; Kimberly LP Long, Stella Chen, Aline Halliday, Investigation; Justin W Kenney, Anne L Wheeler, Formal analysis, Methodology; Paul Frankland, Supervision, Methodology; Carrie Shilyansky, Validation, Investigation, Visualization, Methodology; Karl Deisseroth, Resources, Software, Supervision, Methodology; Dacher Keltner, Resources, Supervision, Funding acquisition; Daniela Kaufer, Resources, Supervision, Funding acquisition, Validation, Methodology, Writing - review and editing

### Author ORCIDs
Inbal Ben-Ami Bartal  https://orcid.org/0000-0001-6823-2770
Jocelyn M Breton  https://orcid.org/0000-0003-0981-1451
Stella Chen  http://orcid.org/0000-0002-9806-5114
Justin W Kenney  http://orcid.org/0000-0001-8790-5184
Paul Frankland  https://orcid.org/0000-0002-1395-3586
Daniela Kaufer  https://orcid.org/0000-0002-3830-5999

## Ethics

Animal experimentation: Rat studies were performed in accordance with protocols approved by the Institutional Animal Care and Use Committee at the University of California, Berkeley. protocol number AUP-2014-11-6943.

## Decision letter and Author response

Decision letter https://doi.org/10.7554/eLife.65582.sa1
Author response https://doi.org/10.7554/eLife.65582.sa2

# Additional files

## Supplementary files

- Supplementary file 1. Detailed list of brain regions used in the figures.

- Supplementary file 2. Means and confidence intervals for brain-wide c-Fos numbers across conditions.

- Supplementary file 3. Means and confidence intervals for c-Fos numbers per brain region.

- Supplementary file 4. Set of parameters used in a series of multinomial regression tests. Combinations of clustering algorithms, parameters, and subsets of data used in *Figure 3D*. 'Cluster group' was used to obtain the cluster assignments for each brain region. 'Reference group' was the control group in the multinomial logistic regression. The 'weighted' column indicates whether weights (correlations) between brain regions were taken into account while performing clustering. 'All': all groups; 'rest': all conditions except ingroup and outgroup; 'baseline': untested baseline condition. The thresholds of percentile rank, p-value, and absolute r value in the parameter column were applied to the covariance matrix of the c-Fos data in the cluster group before feeding it into the selected clustering algorithm.

- Transparent reporting form

## Data availability

Data of c-Fos numbers, calcium signal and movement are uploaded on the Open Science Framework depository (https://osf.io/5qtcx/).

The following dataset was generated:

| Author(s) | Year | Dataset title | Dataset URL | Database and Identifier |
|-----------|------|---------------|-------------|-------------------------|
| Bartal B-AI | 2021 | Neural correlates of ingroup bias for pro-sociality in rats | https://osf.io/5qtcx/ | OSF, osf.io/5qtcx/ |

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
