## [Decision Letter]

**Acceptance summary:**

This manuscript provides a comprehensive combination of behavioral, cellular, and circuit-based analyses that support a role for the nucleus accumbens in social motivation. The results provide novel insight regarding the neural mechanisms governing prosociality, and will therefore be of broad interest to the scientific community.

**Decision letter after peer review:**

Thank you for submitting your article "Neural Correlates of Ingroup Bias for Pro-sociality in Rats" for consideration by *eLife*. Your article has been reviewed by 3 peer reviewers, and the evaluation has been overseen by a Reviewing Editor and Kate Wassum as the Senior Editor. The reviewers have opted to remain anonymous.

Essential Revisions:

In addition to the reviewers' detailed comments below, the editors and reviewers have agreed that the following essential revisions must be addressed:

1) A more rigorous analysis of behavioral data (as outlined in the reviewers' comments below).

2) An enhanced description of the cFos data and calcium transients as they relate to behavior.

3) Further consideration of the natural variation inherent in the reported behaviors.

4) Inclusion of data on cell counts and cell density, as well as representative Fos images that generated those data.

5) Reconsideration of the terminology "ingroup" vs. "outgroup".

6) While we agreed that the addition of causal experiments could help expose which (if any) network is critical for helping behavior, we also agreed that the discovery driven nature of these experiments has high value and that such additional causal experiments are not crucial. Rather we think the revision could be reframed as a discovery-based, exploratory investigation and should include a stronger discussion of the correlational nature of the results, putting these correlational observations within the context of what is already known about the NAc in social reward in the discussion.

Please also include a key resource table, if you have not already done so.

*Reviewer #1:*

The submission by Ben-Ami Bartal et al., reported the neural correlates of helping behavior in rats. The authors trained rats to release a peer from the same strain (ingroup) or different strain (outgroup). Then the authors performed a brain-wide neural activity analysis, using Fos as a neuronal activity marker. The behavioral results showed that rats helped ingroup, but not outgroup members, by releasing them from a restrainer. The Fos data identified a shared network including frontal and insular cortices, active independently of group membership. Interestingly, the striatum region was selectively active for ingroup rats. The author further identified the nucleus accumbens (NAc) as a central network hub correlated with the releasing a peer behavior. Additionally, in vivo imaging investigation revealed NAc activity when a rat was approaching a trapped ingroup member. Finally, by using a retrograde tracer plus Fos, the authors identified a subpopulation of activated anterior cingulate cortex  NAc projection correlated with helping behavior in the ingroup rats. Overall, the manuscript provides informative data on the neural correlates of helping behavior providing some inputs for mechanistic investigations of this phenomenon.

The main limitation is that at its current stage, the work is just descriptive and includes only correlational analyses. While the network brain-wide approach is intriguing, it is not clear which network (if any) is critical for helping behavior (relative to the other controls reported here). These are critical experiments for this type of studies. The exclusion of female subjects from the study significantly decreases the impact of the study and its generalizability. It is difficult to reconcile the in vivo data with Fos. For Fos it is difficult to identify the increased activity during the approaching of a trapped ingroup member, just because the timeline of Fos expression is extremely different than the in vivo investigation.

1. The main limitation is that at its current stage, the work is just descriptive and includes only correlational analyses. While the network brain-wide approach is intriguing, it is not clear which network (if any) is critical for helping behavior (relative to the other controls reported here). These are critical experiments for this type of studies.

2. The exclusion of female subjects from the study significantly decreases the impact of the study and its generalizability.

3. It is difficult to reconcile the in vivo data with Fos. For Fos it is difficult to identify the increased activity during the approaching of a trapped ingroup member, just because the timeline of Fos expression is extremely different than the in vivo investigation. The author should consider to either remove the in vivo data or provide a rationale/justification for this technical concern.

4. The retrograde tracing experiment seems to be disconnected from the rest of the paper and, as reported by the authors, under developed. Initially the investigation is about the comparison between ingroup and outgroup (together with the other control groups). Then for the retrograde tracing experiment the authors focused only on the ingroup rats. This is confusing because the retrograde tracing experiment should be useful to confirm the networks identified with the Fos analysis.

5. Page 7 refers to Figure 2G but this figure is missing in the panel.

*Reviewer #2:*

In their study "neural correlates of ingroup bias for pro-sociality in rats", the authors conduct analyses of IEG induction and calcium imaging during a helping behavior test (HBT) to characterize networks of core brain regions that are functionally responsive to helping behavior towards "ingroup" vs. "outgroup" rats. Rats demonstrate a bias towards helping trapped individuals of the same strain (ingroup) compared to releasing rats of a different strain (outgroup). The results identify a collection of functionally correlated regions that appear to be selectively activated under ingroup, but not outgroup, HBT conditions. Of these regions, the nucleus accumbens arises as a central node whose cFos induction and calcium transients' correlate with approaching a trapped ingroup conspecific. The authors argue that these novel results highlight NAc activity as a neural correlate of ingroup bias and may serve as a substrate mediating basic empathic responses in a lower vertebrate. The authors further suggest that such network activity may reflect evolutionarily conserved mechanisms that give rise to more complex empathic, and at times maladaptively biased, social phenotypes in humans.

The authors sampled an impressive number of brain regions for their cFos analyses, and the convergence with the fiber photometry data provided compelling evidence that the NAc is indeed involved in these complex prosocial behaviors. The findings are novel, if correlative, and the behavioral paradigm used in conjunction with calcium imaging techniques provide a rich opportunity to understand the neural mechanisms governing prosociality, which is of broad interest to the scientific community.

The experimental paradigm provides a rich behavioral context to examine the neural correlates of helping behavior. However, collapsing the behavioral dependent variable into a binary measure of opening vs. not opening, or using just the percentage of rats that opened, ignores the potentially nuanced nature of these prosocial behaviors and their neural substrates. Quantifying what the free rats (and trapped rats) were doing over the course of the HBT would help more rigorously contextualize what the cFos data correspond to at a finer behavioral scale. Such an analysis is also needed to help to evaluate whether the trapped mouse is in distress after 12 days of training (habituation), which is critical to the empathic response explanation for the corresponding neural data. I would caution against the authors' use of ingroup/outgroup terminology as a framework for the experiments as it invokes potentially anthropocentric interpretations and explanations of the rats' behavior. As these terms emerge from the social psychology literature and typically regard cultural processes underlying social identity, there is not enough evidence that parallel cognitive processes are driving the behavioral phenotypes in the rats.

1. After 12 days of training, in which the trapped rat appears to be trapped for at least 20 mins for the ingroup condition, can we safely assume that the paradigm is still distressing for the trapped rat? The empathic response explanation for the corresponding neural data is contingent upon the notion that the rat in the trap is experiencing distress. As the trapped rat becomes more habituated to being trapped, does this pose as a confound for the authors interpretations? Does the trapped rat re-enter the trap after release? This could help characterize whether or not the trap was still negatively valenced.

2. The experimental paradigm provides a rich behavioral context to examine the neural correlates of helping behavior. However, collapsing the behavioral dependent variable into a binary measure of opening vs. not opening, or using just the percentage of rats that opened, ignores the potentially nuanced nature of these prosocial behaviors and their neural substrates. Quantifying what the free rats (and trapped rats) were doing over the course of the HBT would help more rigorously contextualize what the cFos data correspond to at a finer behavioral scale. Are they still trying to open the restrainer throughout? Do they give up part way? The cFos totals and calcium transients relate to a combination of the experimental conditions and the subject's behavior during the HBT. Without extensive behavioral analysis it's not clear what the neural activity truly reflects (i.e. internal motivation to open the previously openable trap? A response to whatever signals are being emitted from the trapped conspecific? Etc. ). One could also correlate or regress number of opening attempts with the cFos counts to see if there's a significant relationship between them.

3. Further, there is very interesting natural variation in the propensity of rats to learn to become "openers". A non-trivial number of free rats did not seem to become openers, but the implications of these outcomes were not discussed. Were the 2/8 rats that failed to become openers in the ingroup condition included in the cFos analyses? Or the 5/13 in the retrograde cFos analyses? Combining these would result in a reasonable sample size for analysis. Incorporating the status of the free rat as an opener or non-opener may be critical to interpreting the neural data.

4. I would caution against the authors' use of ingroup/outgroup terminology as a framework for the experiments as it invokes potentially anthropocentric interpretations and explanations of the rats' behavior. As these terms emerge from the social psychology literature and typically regard cultural processes underlying social identity, there is not enough evidence that parallel cognitive processes are driving the behavioral phenotypes in the rats. The authors included the phrase "social selectivity", which I believe better captures the phenomenon without invoking unnecessary higher order associations. There are other concerns with using ingroup/outgroup to refer to strain and how this may be construed with respect to ethnicity/race.

5. Is the outgroup trapped rat always the same animal? Differences in stimulus ID exposure during training for the HBT may mean cFos responses are tied to stimulus familiarity.

6. Line 382- are these boldness results available? Or the OFT results? And why run a test of exploration anxiety-like behavior in an arena in which the animals have been habituated? Why run these tests and not report the results?

7. Line 139 – The brief condition is used to isolate the neural response of helping vs. exposure to a trapped rat, but the differences in the experiences of the trapped rat (12 days, 1 completely trapped vs. 3 days completely trapped) could also contribute to the variation in neural activity of the free rat that is not easily decomposed into a helping response + noise.

8. Line 155- The baseline condition rats were presumably not handled, they remained in their home cage instead of moving to an arena and were not exposed to a trap. Therefore, it's not entirely clear what experimental components the differential neural responses are related to, and this should be mentioned.

9. Line 252 – It's curious that activity would decrease, given that other studies show significant increases in NAc activity upon reward consumption (water, sucrose, social interaction) and this not what is observed upon acquisition of social reward when opening the restrainer. A short discussion of this discrepancy is warranted.

10. Differences in locomotion could contribute to the differences in NAc Ca^2+^ activity. The authors partially address this in line 320, but this explanation does not take into account potential differences in velocity upon entry to the area around the restrainer – a particular concern given that differences in velocity are noted in Figure 1: "The measure of entry into the area around the restrainer used in the fiber photometry experiment by definition controls for movement, as it depicts the same movement across conditions as the event of interest. Thus, while motor processing is part of the neural processing displayed by these rats, it is unlikely to be the primary explanation for the differences observed between groups."

11. Ca^2+^ signals in the NAc commonly exhibit attenuation over time as novelty seems to be a strong driver of activity. This can lead to false interpretations of the data if the behavior is not balanced across groups/condition. Did the animals show equivalent number of approach bouts for in- and outgroup members? If so, were there significant changes in fluorescence across approach bouts?

*Reviewer #3:*

The manuscript under review, by Bartal et al. titled "Neural Correlates of Ingroup Bias for Pro-Sociality in Rats" provides a comprehensive combination of behavioral, cellular, and circuit-based analysis. Briefly, the authors first show that male rats have an "ingroup" bias for pro-social behavior using their previously introduced helping behavior task (HBT). Specifically, rats preferred to help trapped rats of the same (ingroup), but not different (outgroup), strain. Using this procedure, in conjunction with an exhaustive set of control conditions(!), the authors next performed Fos activity mapping in 45 brain regions spanning bregma +4.2mm to -5.64mm. The statistical approaches for this Fos mapping experiment were Partial Least Square analysis with permutation and boot strapping tests. The data show that HBT conditions are overall elevated in contrast comparted to controls, with the ingroup condition greater than the outgroup condition. Subsequent analysis of only the ingroup vs outgroup conditions compared to base-line non-tested rats shows that there are brain regions in both conditions that are commonly or uniquely activated by either HBT condition. Of specific interest, the MO, PrL, NAc and LS were significantly more active in the ingroup than outgroup condition. These differences were not found in a group of rats trained to retrieve chocolate pellets from the HBT apparatus (although few rats acquired this task). The authors propose this neural correlate therefore is specific to social reward. A second set of analysis were then performed on the same datasets using network graph analysis, which further refined the list of brain regions of interest to the NAc. To determine how the NAc encodes HBT behavior, the authors used non-conditional GCaMP in the NAc in conjunction with fiber photometry, revealing the NAc is most active during the HBT within the ingroup condition. Lastly, the authors use retrograde tracing to show that a sub-population of NAc projecting ACC neurons are activated by pro-social approach.

The authors do an excellent job of highlighting the potential methodological and interpretational caveats of their results within the main text, as well as detail unexpected methodological issues within the methods section. Thank you for this transparency. Below I provide comments regarding these caveats.

1) It appears there is very little behavioral plasticity between the ingroup and outgroup rats. That is, across 2 weeks of repeated testing none of the outgroup rats transition to receiving empathy-like behavior from their partner. From the methods, it is unclear to me what the housing conditions are with the ingroup and outgroup conditions. I think the housing is that ingroup rats are pair-housed with their partner for the duration of the experiment, and outgroup rats are introduced to their (consistently same) partner during the 1 hour test periods? Is this correct or incorrect? Regardless, please clarify the housing conditions for ingroup and outgroup conditions across the full experimental time-line.

2) As a follow-up, since I am not sure of the housing conditions, could the authors speculate on if there is behavioral plasticity for empathy in rats and if they can transition between "ingroup" and "outgroup" conditions? How much exposure does this require, or does it never happen? Since I am unsure of the exact methods, I am curios if this ingroup bias is more a result of the methods (familiarity with a housed partner vs being a rat of another strain). If a Sprague-Dawley was housed with a Long-Evans cage-mate as their ingroup, would the ingroup bias still exist? In rats, is ingroup/outgroup a function of social familiarity or genetic background?

3) This is a herculean fos mapping effort. The authors provide cell count data but gloss over analysis of cell counts or cell density, jumping directly to more complex PLS and bootstrapping. It is unclear to me of any brain regions, as group means, are significantly different in Fos+ cell counts between HBT conditions.

4) I am not an expert in PLS and bootstrapping analysis of fos mapping data, so I cannot speak directly these results. However, I would like to see representative Fos images of the brain regions that were identified by these approaches as a main or supplemental figure, between conditions. I believe these would be the regions referenced in Figure 2F.

6) Similar to point 4, it would be helpful if the authors provided representative Fos images of the key brain regions identified in by graph network theory. Since some of these overlap with the PLS analysis, perhaps a figure could be provided that has the relevant Fos images between conditions from these two analysis?

7) The NAc has a long and prolific literature surroundings its role in various social reward behaviors, most recently highlighted by work in the stress anhedonia and aggression reward fields identifying cell-type specific roles for NAc medium spiny neurons in controlling social reward-related behaviors. The photometry data is very compelling. Can the author speculate on the NAc cell types that are underlying their photometry observations?

8) The NAc has been the focus of intense scrutiny across numerous social behaviors, ranging from stress-related social anhedonia to sexual motivation to aggression motivation. The discussion would notably strengthen by relating the current results within the context of previously identified roles for the NAc in other forms of social motivation behaviors.

Overall, the authors have presented a well-crafted series of experiments that are appropriately controlled.

5) There is no Figure 2G, referenced in line 156.

[Editors' note: further revisions were suggested prior to acceptance, as described below.]

Thank you for resubmitting your work entitled "Neural Correlates of Ingroup Bias for Pro-sociality in Rats" for further consideration by *eLife*. Your revised article has been evaluated by Kate Wassum (Senior Editor) and a Reviewing Editor.

The manuscript has been improved but there are some remaining issues that need to be addressed, as outlined below:

The concerns regarding the original submission have been addressed. The revised manuscript provides novel insight regarding the role of the nucleus accumbens and surrounding circuitry in prosociality, and will therefore be of broad interest to the scientific community. The reviewers have a few remaining concerns regarding the Figures that need to be addressed:

1) The representative cFos image shown in Figure 2B should be replaced with a cFos image from the nucleus accumbens, as this is the brain region that was used for recording.

2) To better illustrate the co-labels with cFos in the merged image in Figure 5D, the authors should consider pseudo-coloring the flurogold label.

---

## [Author Response]

Essential Revisions:In addition to the reviewers' detailed comments below, the editors and reviewers have agreed that the following essential revisions must be addressed:1) A more rigorous analysis of behavioral data (as outlined in the reviewers' comments below).

An in-depth analysis of movement pattern of rats along the session days and minutes has been added, in order to better understand the rats’ motivational state during the helping test. The added analyses are detailed below. Broadly, we found that:

While rats were generally more motivated to move and act around a trapped ingroup member than an outgroup member along the 12 days of testing in the helping behavior test, differences in motor activity were not significant on the cFos day itself. Rats did spend more time near the restrainer, suggesting that motivational state rather than velocity was associated with increased c-Fos activity. In the photometry experiment, an analysis of the velocity as rats entered the area around the restrainer found no differences for the ingroup and outgroup sessions, providing further evidence that differences in the recorded calcium signal represent primarily the social context.

2) An enhanced description of the cFos data and calcium transients as they relate to behavior.

Several figures have been added to provide insight into the association between c-Fos levels and movement data, in a new figure (Figure 2 —figure supplement 5 in the current manuscript). An analysis of movement in the photometry experiment has been added both as summary statistics in the results and an additional graph (Figure 4G).

3) Further consideration of the natural variation inherent in the reported behaviors.

A new analysis has been added comparing neural activity in openers and non-openers in the retrograde experiment (see new Figure 6). Additionally, individual variation can be observed across the manuscript in figure 1G-H; Figure 2 —figure supplement 3, 4C-E, and 5.

4) Inclusion of data on cell counts and cell density, as well as representative Fos images that generated those data.

Data on cell counts is provided on the public repository OSF, a visual presentation in Figure 2 —figure supplement 3, and a new figure presenting representative c-Fos images has been added to the supplementary methods (new Figure 2 —figure supplement 1B).

5) Reconsideration of the terminology "ingroup" vs. "outgroup".

A detailed discussion of this concern is provided below, in the detailed response to the reviewers’ comments. In brief, the terms ‘ingroup’ and ‘outgroup’ are commonly used in the sociobiology literature to refer to non-human social groups and to describe behavior between these groups. However, to emphasize the difference between this meaning and the more cultural usage in human social psychology, a clearer definition has been added to the introduction, including the limitation of this terminology in comparison with human behavior. The terms are now introduced in parenthesis, followed by the operational definition.

Furthermore, it’s worth noting that these terms are used in particular based on previous results demonstrating that rats (a) determine pro-social behavior based on group identity rather than individual familiarity and (b) these groups are flexible and determined by social experience (Bartal et al., *eLife* 2014). In this manuscript, the ‘ingroup’ members included both cage-mates and strangers of the same strain, rendering either the term ‘strain’ or ‘familiarity’ partially accurate for describing the relationship between the free and trapped rat. Thus, despite the obvious differences between rats and humans, we request to be allowed to retain these terms, as they best reflect the effect we observe.

6) While we agreed that the addition of causal experiments could help expose which (if any) network is critical for helping behavior, we also agreed that the discovery driven nature of these experiments has high value and that such additional causal experiments are not crucial. Rather we think the revision could be reframed as a discovery-based, exploratory investigation and should include a stronger discussion of the correlational nature of the results, putting these correlational observations within the context of what is already known about the NAc in social reward in the discussion.

We are in agreement. The abstract has been modified accordingly, and the following text has been added to the introduction:

“Here we employed a discovery-driven approach to compare the brain-wide activation pattern in response to trapped ingroup and outgroup members, following the helping behavior test. This investigation, which aimed at providing a broad and unbiased overview, led to the identification of central hubs specifically active during the ingroup condition, where rats demonstrated prosocial intent.”

Furthermore, the following discussion of the correlational nature of the manuscript has been added to the discussion:

“The results presented above are correlational in nature and descriptive only. Correlation data in and of itself is not sufficient evidence, especially given the limited sample size and possibility of false discoveries. Thus, these data provide a partial picture of the rats’ neural response and should be interpreted with this caveat in mind. It is nonetheless encouraging that the Nac emerged as a key region across several measures. The increase in c-Fos^+^ cells observed in the ingroup condition was mirrored by the in-vivo calcium signal, and both measures were associated with behavior (helping and approach respectively)…Overall, this manuscript aims to provide a broad overview of the neural circuitry associated with pro-social intent and provides a base for future work that will aim to dissect these circuits and understand their causal contribution to behavior.”

Please also include a key resource table, if you have not already done so.

A key resource table has been added to the methods.

Reviewer #1:The submission by Ben-Ami Bartal et al., reported the neural correlates of helping behavior in rats. The authors trained rats to release a peer from the same strain (ingroup) or different strain (outgroup). Then the authors performed a brain-wide neural activity analysis, using Fos as a neuronal activity marker. The behavioral results showed that rats helped ingroup, but not outgroup members, by releasing them from a restrainer. The Fos data identified a shared network including frontal and insular cortices, active independently of group membership. Interestingly, the striatum region was selectively active for ingroup rats. The author further identified the nucleus accumbens (NAc) as a central network hub correlated with the releasing a peer behavior. Additionally, in vivo imaging investigation revealed NAc activity when a rat was approaching a trapped ingroup member. Finally, by using a retrograde tracer plus Fos, the authors identified a subpopulation of activated anterior cingulate cortex  NAc projection correlated with helping behavior in the ingroup rats. Overall, the manuscript provides informative data on the neural correlates of helping behavior providing some inputs for mechanistic investigations of this phenomenon.The main limitation is that at its current stage, the work is just descriptive and includes only correlational analyses. While the network brain-wide approach is intriguing, it is not clear which network (if any) is critical for helping behavior (relative to the other controls reported here). These are critical experiments for this type of studies. The exclusion of female subjects from the study significantly decreases the impact of the study and its generalizability. It is difficult to reconcile the in vivo data with Fos. For Fos it is difficult to identify the increased activity during the approaching of a trapped ingroup member, just because the timeline of Fos expression is extremely different than the in vivo investigation.1. The main limitation is that at its current stage, the work is just descriptive and includes only correlational analyses. While the network brain-wide approach is intriguing, it is not clear which network (if any) is critical for helping behavior (relative to the other controls reported here). These are critical experiments for this type of studies.2. The exclusion of female subjects from the study significantly decreases the impact of the study and its generalizability.

We agree. Experiments with female rats are currently underway. We have added the following text to the discussion:

“It is important to note that as the data presented here represents activity in male rats only, conclusions are limited to males. The collection of data from female rats is currently ongoing and will be important for generalizing these results.”

3. It is difficult to reconcile the in vivo data with Fos. For Fos it is difficult to identify the increased activity during the approaching of a trapped ingroup member, just because the timeline of Fos expression is extremely different than the in vivo investigation. The author should consider to either remove the in vivo data or provide a rationale/justification for this technical concern.

The advantage of using both the c-Fos and photometry data together is to provide converging evidence from two different methods with different limitations. As noted, c-Fos provides low temporal resolution but high spatial resolution, whereas the photometry provides high temporal resolution but low spatial resolution. These two methods are meant to complement one another. cFos is used to contrast the behavioral experience over the entire session between the ingroup and outgroup conditions. Photometry provides a time series signal of activity change along the task, and provides an opportunity to compare the different conditions within subject. The two datasets propose to capture different aspects of the neural response, and provide supporting evidence for the role of the Nac in the helping test.

4. The retrograde tracing experiment seems to be disconnected from the rest of the paper and, as reported by the authors, under developed. Initially the investigation is about the comparison between ingroup and outgroup (together with the other control groups). Then for the retrograde tracing experiment the authors focused only on the ingroup rats. This is confusing because the retrograde tracing experiment should be useful to confirm the networks identified with the Fos analysis.

The rationale behind the retrograde experiment was to test which structural projections may be playing a role in pro-social approach. The network data derived from the c-Fos experiments provide information about the functional connectivity, and here we are investigating structural connectivity. The specific question answered in this experiment was which structural projections into the Nac are also active during the helping test, as determined by c-Fos+ cells. As the other findings suggested the Nac is a central hub in the ingroup network, this region was chosen for this experiment. We agree that adding a condition with the outgroup would be informative. However, as there is no helping in the outgroup, and the Nac is not active according to our findings, this comparison would not necessarily add useful information. This experiment provides a lead for future investigation narrowing in on structural projections that are necessary for helping to occur.

5. Page 7 refers to Figure 2G but this figure is missing in the panel.

Fixed.

Reviewer #2:In their study "neural correlates of ingroup bias for pro-sociality in rats", the authors conduct analyses of IEG induction and calcium imaging during a helping behavior test (HBT) to characterize networks of core brain regions that are functionally responsive to helping behavior towards "ingroup" vs. "outgroup" rats. Rats demonstrate a bias towards helping trapped individuals of the same strain (ingroup) compared to releasing rats of a different strain (outgroup). The results identify a collection of functionally correlated regions that appear to be selectively activated under ingroup, but not outgroup, HBT conditions. Of these regions, the nucleus accumbens arises as a central node whose cFos induction and calcium transients' correlate with approaching a trapped ingroup conspecific. The authors argue that these novel results highlight NAc activity as a neural correlate of ingroup bias and may serve as a substrate mediating basic empathic responses in a lower vertebrate. The authors further suggest that such network activity may reflect evolutionarily conserved mechanisms that give rise to more complex empathic, and at times maladaptively biased, social phenotypes in humans.The authors sampled an impressive number of brain regions for their cFos analyses, and the convergence with the fiber photometry data provided compelling evidence that the NAc is indeed involved in these complex prosocial behaviors. The findings are novel, if correlative, and the behavioral paradigm used in conjunction with calcium imaging techniques provide a rich opportunity to understand the neural mechanisms governing prosociality, which is of broad interest to the scientific community.The experimental paradigm provides a rich behavioral context to examine the neural correlates of helping behavior. However, collapsing the behavioral dependent variable into a binary measure of opening vs. not opening, or using just the percentage of rats that opened, ignores the potentially nuanced nature of these prosocial behaviors and their neural substrates. Quantifying what the free rats (and trapped rats) were doing over the course of the HBT would help more rigorously contextualize what the cFos data correspond to at a finer behavioral scale.

This is right. Perhaps because of the previous emphasis given to behavior in the last publications, the behavioral analysis was somewhat lean in this manuscript. We have now added a thorough analysis of the movement patterns, including a new figure (Figure 2 —figure supplement 4) and new analysis for the photometry movement data. These analyses provide insight about the capacity of a trapped ingroup member to motivate rats to persist in the typical pattern of circling the restrainer containing the trapped rat and attempting to reach the rat inside, or release him, that was previously described in Ben-Ami Bartal, 2011, 2014, 2016.

Such an analysis is also needed to help to evaluate whether the trapped mouse is in distress after 12 days of training (habituation), which is critical to the empathic response explanation for the corresponding neural data.

While this dataset does not inform the question of whether the animals were still stressed at the end of testing, corticosterone levels were tested in a previous manuscript in 50 pairs of rats during the first and last days of the helping test. Then it was found that both the free and trapped rats demonstrated an increase in cort levels compared to baseline and compared to control rats on the first days of testing. In opener pairs this response was abolished by the last day of testing, but in non-opener pairs it remained high, probably since rats remained trapped. (this data is only partially published, in Ben-Ami Bartal et al., Front. Psych, 2016) In the current experiments, it is likely that rats were similarly stressed on the final day. Even opener pairs, as the restrainer was latched, and no one was released. While a direct test of this remains to be done, please consider as well that restraint is a highly validated stressor and has been used for chronic stress paradigms, where classically rats are trapped daily for 2-3 weeks. Habituation likely does not reduce stress in this context. Habituation is effective in reducing stress when animals learn the new situation is not aversive. As rats experience restraint as aversive, they continue to be negatively aroused across testing. The following text has been added to the discussion:

“Undoubtedly, social reward is an important driver of mammalian behavior, and rats experience social reward during this paradigm. However, the helping test is a stressful situation. Restraint is a well validated chronic stressor (Glavin et al., 1994; Ottenweller, Servatius, and Natelson, 1994; Pare and Glavin, 1986; Servatius et al., 2000), and testing induces a stress response in both the free and the trapped rat, including secretion of the stress hormone corticosterone, defecation and freezing(Ben-Ami et al., 2011; Ben-Ami et al., 2016). It is thus important to consider that the behavior of the free rat is influenced by the stress of the trapped rat.”

I would caution against the authors' use of ingroup/outgroup terminology as a framework for the experiments as it invokes potentially anthropocentric interpretations and explanations of the rats' behavior. As these terms emerge from the social psychology literature and typically regard cultural processes underlying social identity, there is not enough evidence that parallel cognitive processes are driving the behavioral phenotypes in the rats.

While we agree that the terms ‘ingroup’ and ‘outgroup’ have been typically used to describe a cultural construct rather than a biological one, we would argue that the meaning of this terms best captures the animals’ behavior. As we have previously demonstrated, rats help others who belong to a certain group (strain) based on their previous social experience with other members of this group. As such, Sprague-Daweley rats fostered at birth with Long-evans litters help only members (even strangers) of the adoptive strain as adults, and do not help strangers of their own strain. Even as adults, after 2 weeks of co-housing with a Long-evans cage-mate, rats help Long-evans strangers (Ben-Ami Bartal et al., *eLife* 2014). This finding shows that pro-social behavior is determined according to experience with a member of the strain, but not the biological identity of the trapped rat. Thus pro-social behavior is flexible, and determined by the experience with the group, rather than individual familiarity, or biological similarity. This behavior is similar to ingroup bias in humans, who show a preference for ingroup members even if the ingroup is determined by minimal cues. Furthermore, a diverse environment (e.g. in school) has been shown to reduce ingroup bias in humans and increases empathic responding to members of other ethnic groups experiencing pain. As rats will release Long-evans strangers following co-housing with a long-evans cage-mate, it would be inaccurate to use neither ‘strain’ nor ‘familiarity’ as the critical factor determining the behavioral phenotype. Rather, it is a “strain with which the rat is familiar” and, we suspect, has had positive social experience with (has bonded with) a member of the strain. In-group bias is a term used to describe a preference for pro-social behavior among members of the same group, where groups can be determined by innate characteristics such as sex, or formed by the environment. Usage of the term “ingroup bias” or “ingroup favoritism” to describe social behavior in non-human animals is frequent and accepted. Ingroup bias has been widely documented both in humans and non-human animals, from ants to dolphins and birds. The terms ingroup and outgroup are now first presented in parentheses (“ingroup” and “outgroup”) followed with an operational definition. The following has been added to the introduction:

“Importantly, rats demonstrate an ingroup bias, releasing “ingroup members” (rats of the same strain, whether familiar or not), but not “outgroup members” (rats of an unfamiliar strain).”

“The terms “ingroup” and “outgroup”, while often used in regard to cultural processes underlying social identity in humans, are also used for describing socially-selective affiliation in non-human animals (Masuda and Fu, 2015; Nakamura and Masuda, 2012; Robinson and Barker, 2017), and are adopted here, despite likely differences in their neurobiological structure across species.”

1. After 12 days of training, in which the trapped rat appears to be trapped for at least 20 mins for the ingroup condition, can we safely assume that the paradigm is still distressing for the trapped rat? The empathic response explanation for the corresponding neural data is contingent upon the notion that the rat in the trap is experiencing distress. As the trapped rat becomes more habituated to being trapped, does this pose as a confound for the authors interpretations? Does the trapped rat re-enter the trap after release? This could help characterize whether or not the trap was still negatively valenced.

See answer for this as above.

Regarding the point of re-entry into the restrainer, it is a commonly observed behavior. However, it does not provide evidence that being trapped is not aversive. On the contrary, It’s likely the restrainer itself is appealing for being a small protected place. But when trapped inside, it becomes aversive. This is evidenced by attempts to escape, defecations, and occasional breakouts, especially as the experiment progresses. Furthermore, when the door was turned around in previous unpublished experiments, trapped rats quickly released themselves. The analogy might be to sitting comfortably in your room, but if suddenly you were to discover the door was locked from the outside, you’d start feeling uncomfortable and try to find a way out. This does not mean that you find the room itself aversive, it is the situation of being locked inside that is stressful.

2. The experimental paradigm provides a rich behavioral context to examine the neural correlates of helping behavior. However, collapsing the behavioral dependent variable into a binary measure of opening vs. not opening, or using just the percentage of rats that opened, ignores the potentially nuanced nature of these prosocial behaviors and their neural substrates. Quantifying what the free rats (and trapped rats) were doing over the course of the HBT would help more rigorously contextualize what the cFos data correspond to at a finer behavioral scale. Are they still trying to open the restrainer throughout? Do they give up part way? The cFos totals and calcium transients relate to a combination of the experimental conditions and the subject's behavior during the HBT. Without extensive behavioral analysis it's not clear what the neural activity truly reflects (i.e. internal motivation to open the previously openable trap? A response to whatever signals are being emitted from the trapped conspecific? Etc. ). One could also correlate or regress number of opening attempts with the cFos counts to see if there's a significant relationship between them.

We have added an extensive analysis in response to this point. First, an analysis of movement pattern along the 12 testing days of the HBT. This is presented in Figure 1. Next, an analysis of movement on the c-Fos day, which can be associated with the c-Fos levels. This analysis shows the activity levels and location near the restrainer of the free rats, indicating the motivational state was more persistent in the ingroup condition. This is marked by a consistently high amount of time spent near the restrainer on the later days and times in the session by the free rat in the ingroup condition, whereas this approach decreases as time goes by in the outgroup condition. This finding also lines up with previously published data showing an increased ratio of activity in the first and second halves of the session compared to control conditions (Ben-Ami Bartal, 2011). On the c-Fos day, the movement differences were not significant, yet there was a significant correlation between time spent around the restrainer and brain-wide c-Fos levels, indicating that neural activity was higher the more rats remained near the trapped rat. while we did not conduct a manual coding of opening attempts (pretty tricky to do this in a standard, objective way, working on it!), this measure would be a derivative of time near the restrainer, as rats would have to be in the area around the restrainer to attempt door-opening.

3. Further, there is very interesting natural variation in the propensity of rats to learn to become "openers". A non-trivial number of free rats did not seem to become openers, but the implications of these outcomes were not discussed. Were the 2/8 rats that failed to become openers in the ingroup condition included in the cFos analyses? Or the 5/13 in the retrograde cFos analyses? Combining these would result in a reasonable sample size for analysis. Incorporating the status of the free rat as an opener or non-opener may be critical to interpreting the neural data.

In this manuscript, for all original analyses, all animals from the tested condition were included, both openers and non-openers. At your suggestion, a new analysis of openers vs. non-openers has been included (Figure 6). This is based off the 5vs8 comparison in the retrograde experiment. In addition, the two non-openers in the c-Fos experiment are marked in lighter gray in the dot movement graphs in Figure 1.

Historically, 30% of rats tested with ingroup members never become openers. We are very interested in understanding these individual differences. Were the non-openers unable to learn the task? Or perhaps less motivated? From observations, it appears they tend to be of a more anxious profile. As yet, we have not gathered a convincing body of evidence to support either alternative. We are currently working on developing algorithms for a more sophisticated analysis of the rats’ movement data to address these questions. A few differences were identified in this analysis, as detailed in the results, providing interesting leads for future studies with a larger sample. In addition to the new analysis, the following text has been added to the discussion:

“Regarding the motivational state underlying door-opening, consider the following: when rats open the restrainer quickly on consecutive days and become ‘openers’, it is clearly an intentional behavior. However, when rats fail to open the restrainer, it could be explained by either lack of will or lack of ability to open the door. Here, some rats tested with ingroup members did not become ‘openers’. Previous experiments similarly found that while most rats help ingroup members, around 30% of Sprague-Dawley rats don’t become openers. The current dataset does not provide enough information to support any conclusions regarding the motivational state of the ‘non-openers’ to surmise whether they were unable or unwilling to open the restrainer. Analysis presented above suggest the OFC and BLA are potential targets for future investigation of these two behavioral phenotypes. Individual variability of helping behavior is of great interest and needs to be investigated in depth in future studies.”

4. I would caution against the authors' use of ingroup/outgroup terminology as a framework for the experiments as it invokes potentially anthropocentric interpretations and explanations of the rats' behavior. As these terms emerge from the social psychology literature and typically regard cultural processes underlying social identity, there is not enough evidence that parallel cognitive processes are driving the behavioral phenotypes in the rats. The authors included the phrase "social selectivity", which I believe better captures the phenomenon without invoking unnecessary higher order associations. There are other concerns with using ingroup/outgroup to refer to strain and how this may be construed with respect to ethnicity/race.

Please see our response above.

5. Is the outgroup trapped rat always the same animal? Differences in stimulus ID exposure during training for the HBT may mean cFos responses are tied to stimulus familiarity.

This is an excellent question. In the past, I’ve tried both alternating the trapped strangers daily so that the free rat never met the same rat twice (as published in *eLife* 2014), and keeping the same trapped stranger over the entire HBT (unpublished). In both setups, SD rats did not release the trapped LE, leading us to the conclusion that the 1-hr exposure afforded during the helping test is not sufficient to induce a shift in pro-social motivation in the free SD rat (as published in Bartal, 2014). On the other hand, we know that 2 weeks of cohabitation does cause such a shift. keep in mind that it is the motivation towards the group, and not the individual, which changes, such that rats will help stranger LEs after the co-habitation. This suggests the necessary familiarity is with the strain as a category and not with an individual rat. Future studies will need to pinpoint the exact timeframe and hopefully the neurobiological changes that occur to support this pro-social shift. The following has been added to the methods to clarify this point:

“Housing conditions: rats were housed in pairs, with a cage-mate of the same strain, and were allowed 2 weeks to habituate to their cage-mate prior to any testing. In the c-Fos experiment, ingroup members were SD cage-mates. For the photometry experiments, ingroup members were SD strangers. The trapped outgroup member was a different LE stranger daily.”

“To prevent individual familiarity with the outgroup member, strangers were swapped daily. Note that we previously found rats will release trapped strangers of their own strain, and that familiarity with the strain determines pro-social behavior rather than individual familiarity.”

6. Line 382- are these boldness results available? Or the OFT results? And why run a test of exploration anxiety-like behavior in an arena in which the animals have been habituated? Why run these tests and not report the results?

The boldness and OFT were run because historically that’s the protocol we’ve followed for the HBT. While we did not analyze these results for this experiment, we found it appropriate to report that these procedures were conducted before the start of testing in the HBT. The following has been added to the methods:

“The boldness and open-field testing were conducted in order to remain consistent with previous experiments, but were not analyzed and are thus not reported.”

7. Line 139 – The brief condition is used to isolate the neural response of helping vs. exposure to a trapped rat, but the differences in the experiences of the trapped rat (12 days, 1 completely trapped vs. 3 days completely trapped) could also contribute to the variation in neural activity of the free rat that is not easily decomposed into a helping response + noise.

We agree that the situation is not identical. The reasoning behind this control condition is that rats tested in the brief condition have experienced witnessing a trapped rat, but they don’t have the knowledge that the restrainer can be opened. For them it is merely witnessing a trapped rat. We chose 3 days instead of a single session in order to allow some habituation to the experimental setup. But the c-Fos day reflects for all rats neural activity on that day only, in which rats witness a trapped rat for the entire session. We agree that this control on its own does not purely isolate the response to a trapped animal, as these rats do have different previous experience that influences their response on that day. However, we are contrasting the 12 day condition with the 3 day condition for ingroup vs. outgroup, which does allow a good comparison of the changes between these conditions, and the effect of group membership can thus be dissociated. Another important consideration, note that even in the HBT group rats were usually trapped at least 40 min for several days, before the free rats learned to open the restrainer (and even more so in the HBT outgroup).

8. Line 155- The baseline condition rats were presumably not handled, they remained in their home cage instead of moving to an arena and were not exposed to a trap. Therefore, it's not entirely clear what experimental components the differential neural responses are related to, and this should be mentioned.

Rats in the baseline condition underwent the same habituation as the other conditions. On the day of sacrifice these rats did not undergo behavioral testing. This is now added to the methods. This untested baseline is meant to contrast out unspecific neural activity observed in our c-Fos stains.

Comparing against baseline for both groups tells us also what the exposure to the restrainer/new arena/ experience of the HBT captures above baseline.

9. Line 252 – It's curious that activity would decrease, given that other studies show significant increases in NAc activity upon reward consumption (water, sucrose, social interaction) and this not what is observed upon acquisition of social reward when opening the restrainer. A short discussion of this discrepancy is warranted.

Activity in the Nac has been associated with different aspects of motivated behavior, such as reward consumption, as well as seeking behavior and avoidance of aversive outcome, and relief of pain. The following has been added to the discussion:

“The activation of the Nac observed in this study mirrors a broad literature of the Nac’s role in social reward-related behaviors. The Nac is mainly composed of GABAergic medium spiny neurons (MSNs) that fall into two classes: those expressing primarily dopamine receptor D1 (D1 MSNs) and those expressing primarily dopamine receptor D2 (D2 MSNs). Broadly, these pathways are thought to lead to opposing outcomes, with D1 MSNs and their pathways activated during reward, and D2 MSNs and their pathways activated during aversion (Klawonn and Malenka, 2018) but see (Soares-Cunha et al., 2016). For example, Nac MSNs are associated with stress-induced anhedonia (Bessa et al., 2013), and mice displaying depressive-like behaviors following chronic stress have decreased excitatory input in D1 MSNs and increased excitatory input in D2 MSNs (Francis et al., 2015). Further, increased D1 MSN activity is predictive of animals that are resilient to stress-induced changes in social behavior (Muir et al., 2018). Whereas the experiments presented here do not distinguish Nac cell types, Given the dichotomy of D1 and D2 activity – i.e. that D2 MSNs are typically involved in aversive behaviors, including avoidance (Boschen et al., 2011), and D1 MSNs are typically involved in rewarding behaviors including aggression seeking (Aleyasin et al., 2018; Golden et al., 2019) –we speculate that dopamine receptor D1 MSN cells are involved in prosocial intent. This hypothesis would require further testing and will be an interesting area for future study.”

10. Differences in locomotion could contribute to the differences in NAc Ca^2+^ activity. The authors partially address this in line 320, but this explanation does not take into account potential differences in velocity upon entry to the area around the restrainer – a particular concern given that differences in velocity are noted in Figure 1: "The measure of entry into the area around the restrainer used in the fiber photometry experiment by definition controls for movement, as it depicts the same movement across conditions as the event of interest. Thus, while motor processing is part of the neural processing displayed by these rats, it is unlikely to be the primary explanation for the differences observed between groups."

This analysis has been added to the manuscript, in Figure 4, comparing the velocity of rats upon entry into the zone around the restrainer. No differences were identified. We also ran the same analysis for the c-Fos movement data with a very similar outcome, which strengthens this finding. We did not include it because the c-Fos measure is not temporally sensitive and this is not a very informative measure for that data set, but it did help reassure us of the photometry data. Additional comparisons of velocity and time around the restrainer also revealed no differences in the ingroup and outgroup session compared from the photometry data. The following text has been added to the results:

“As this measure is defined by a specific movement (entry into the zone around the restrainer), the motor movements associated with all events should be similar, and are not a likely cause for the different neural signals across conditions. In evidence of this idea, velocity at the moment of entry into the restrainer was similar for the outgroup session and the following ingroup session (n=8 per group, Figure 4G), and no differences were found in velocity (5.39±0.28; 5.42±0.34 cm\s), number of entries (19.1±1.6; 19.6±3.1), or time spent in the area around the restrainer (198.9±18.7; 223.7±29.1 s) between the ingroup and outgroup sessions respectively (p>0.05, mean ± SEM).”

11. Ca^2+^ signals in the NAc commonly exhibit attenuation over time as novelty seems to be a strong driver of activity. This can lead to false interpretations of the data if the behavior is not balanced across groups/condition. Did the animals show equivalent number of approach bouts for in- and outgroup members? If so, were there significant changes in fluorescence across approach bouts?

No difference was found in the mean number of bouts in the analysis of the ingroup vs outgroup session here (see text above).

Reviewer #3:The manuscript under review, by Bartal et al. titled "Neural Correlates of Ingroup Bias for Pro-Sociality in Rats" provides a comprehensive combination of behavioral, cellular, and circuit-based analysis. Briefly, the authors first show that male rats have an "ingroup" bias for pro-social behavior using their previously introduced helping behavior task (HBT). Specifically, rats preferred to help trapped rats of the same (ingroup), but not different (outgroup), strain. Using this procedure, in conjunction with an exhaustive set of control conditions(!), the authors next performed Fos activity mapping in 45 brain regions spanning bregma +4.2mm to -5.64mm. The statistical approaches for this Fos mapping experiment were Partial Least Square analysis with permutation and boot strapping tests. The data show that HBT conditions are overall elevated in contrast comparted to controls, with the ingroup condition greater than the outgroup condition. Subsequent analysis of only the ingroup vs outgroup conditions compared to base-line non-tested rats shows that there are brain regions in both conditions that are commonly or uniquely activated by either HBT condition. Of specific interest, the MO, PrL, NAc and LS were significantly more active in the ingroup than outgroup condition. These differences were not found in a group of rats trained to retrieve chocolate pellets from the HBT apparatus (although few rats acquired this task). The authors propose this neural correlate therefore is specific to social reward. A second set of analysis were then performed on the same datasets using network graph analysis, which further refined the list of brain regions of interest to the NAc. To determine how the NAc encodes HBT behavior, the authors used non-conditional GCaMP in the NAc in conjunction with fiber photometry, revealing the NAc is most active during the HBT within the ingroup condition. Lastly, the authors use retrograde tracing to show that a sub-population of NAc projecting ACC neurons are activated by pro-social approach.The authors do an excellent job of highlighting the potential methodological and interpretational caveats of their results within the main text, as well as detail unexpected methodological issues within the methods section. Thank you for this transparency. Below I provide comments regarding these caveats.1) It appears there is very little behavioral plasticity between the ingroup and outgroup rats. That is, across 2 weeks of repeated testing none of the outgroup rats transition to receiving empathy-like behavior from their partner. From the methods, it is unclear to me what the housing conditions are with the ingroup and outgroup conditions. I think the housing is that ingroup rats are pair-housed with their partner for the duration of the experiment, and outgroup rats are introduced to their (consistently same) partner during the 1 hour test periods? Is this correct or incorrect? Regardless, please clarify the housing conditions for ingroup and outgroup conditions across the full experimental time-line.

The LE stranger was changed out every day. However, in other experiments (not presented here), we’ve kept the LE stranger’s identity constant over the entire HBT, and did not see a difference in helping behavior. In both setups, SD rats did not release the trapped LE, leading us to the conclusion that the 1hr exposure afforded during the helping test is not sufficient to induce a shift in pro-social motivation in the free SD rat (as published in Bartal, 2014). On the other hand, we know that 2 weeks of co-habitation does cause such a shift. keep in mind that it is the motivation towards the group, and not the individual, which changes, such that rats will help stranger LEs after the cohabitation. This suggests the necessary familiarity is with the strain as a category and not with an individual rat. Future studies will need to pinpoint the exact timeframe and hopefully the neurobiological changes that occur to support this pro-social shift. The following has been added to the methods to clarify this point:

“Housing conditions: rats were housed in pairs, with a cage-mate of the same strain, and were allowed 2 weeks to habituate to their cage-mate prior to any testing. In the c-Fos experiment, ingroup members were SD cage-mates. For the photometry experiments, ingroup members were SD strangers. The trapped outgroup member was a different LE stranger daily.”

“To prevent individual familiarity with the outgroup member, strangers were swapped daily. Note that we previously found rats will release trapped strangers of their own strain, and that familiarity with the strain determines pro-social behavior rather than individual familiarity.”

2) As a follow-up, since I am not sure of the housing conditions, could the authors speculate on if there is behavioral plasticity for empathy in rats and if they can transition between "ingroup" and "outgroup" conditions? How much exposure does this require, or does it never happen? Since I am unsure of the exact methods, I am curios if this ingroup bias is more a result of the methods (familiarity with a housed partner vs being a rat of another strain). If a Sprague-Dawley was housed with a Long-Evans cage-mate as their ingroup, would the ingroup bias still exist? In rats, is ingroup/outgroup a function of social familiarity or genetic background?

The question of group membership (how it’s defined, sensed, modulated) is fascinating. We have previously found that 2 weeks of co-housing will promote helping for the LE cage-mate, and even for LE strangers (*eLife*, 2014). In the same paper, another experiment looked at SD rats fostered at birth with an LE litter, and we found that as adults, these fostered rats helped only strangers of the foster LE strain, not SD strangers. These findings led to the conclusion that pro-social motivation is determined by social experience, rather than the biological identity. In new experiments we will try to identify what neurobiological changes occur over the 2 weeks of co-housing that could account for this difference. It is also our target to understand how group identity is coded and how it modulates activity in the networks associated with empathy and helping.

3) This is a herculean fos mapping effort. The authors provide cell count data but gloss over analysis of cell counts or cell density, jumping directly to more complex PLS and bootstrapping. It is unclear to me of any brain regions, as group means, are significantly different in Fos+ cell counts between HBT conditions.

Thank you! The analysis of a direct comparison of group means is included in the results. Group means were significantly different between HBT conditions. These findings are visualized in Figure 2F and Figure 3, and Supplementary Files 2,3. In Figure 2F, the brain diagrams are based off ANOVAS with the HBT ingroup, HBT outgroup, and baseline used for the (Bonferonni corrected) posthoc comparisons. The four regions identified in Figure 3A are based off a direct comparison of the ingroup and outgroup HBT means. Supplementary File 2 and 3 contain the means, SEMs and CI per condition and brain region.

4) I am not an expert in PLS and bootstrapping analysis of fos mapping data, so I cannot speak directly these results. However, I would like to see representative Fos images of the brain regions that were identified by these approaches as a main or supplemental figure, between conditions. I believe these would be the regions referenced in Figure 2F.

This is now included in Figure 2 —figure supplement 1B. In addition, Figure 2—figure supplement 3 provides a useful visualization of the counts across all test conditions.

6) Similar to point 4, it would be helpful if the authors provided representative Fos images of the key brain regions identified in by graph network theory. Since some of these overlap with the PLS analysis, perhaps a figure could be provided that has the relevant Fos images between conditions from these two analysis?

We included several regions, though not all, in Figure 2 —figure supplement 1B. As there are many interesting areas, it was hard to choose!

7) The NAc has a long and prolific literature surroundings its role in various social reward behaviors, most recently highlighted by work in the stress anhedonia and aggression reward fields identifying cell-type specific roles for NAc medium spiny neurons in controlling social reward-related behaviors. The photometry data is very compelling. Can the author speculate on the NAc cell types that are underlying their photometry observations?

The following has been added to the discussion:

“The activation of the Nac observed in this study mirrors a broad literature of the Nac’s role in social reward-related behaviors. The Nac is mainly composed of GABAergic medium spiny neurons (MSNs) that fall into two classes: those expressing primarily dopamine receptor D1 (D1 MSNs) and those expressing primarily dopamine receptor D2 (D2 MSNs). Broadly, these pathways are thought to lead to opposing outcomes, with D1 MSNs and their pathways activated during reward, and D2 MSNs and their pathways activated during aversion (Klawonn and Malenka, 2018) but see (Soares-Cunha et al., 2016). For example, Nac MSNs are associated with stress-induced anhedonia (Bessa et al., 2013), and mice displaying depressive-like behaviors following chronic stress have decreased excitatory input in D1 MSNs and increased excitatory input in D2 MSNs (Francis et al., 2015). Further, increased D1 MSN activity is predictive of animals that are resilient to stress-induced changes in social behavior (Muir et al., 2018). Whereas the experiments presented here do not distinguish Nac cell types, Given the dichotomy of D1 and D2 activity – i.e. that D2 MSNs are typically involved in aversive behaviors, including avoidance (Boschen et al., 2011), and D1 MSNs are typically involved in rewarding behaviors including aggression seeking (Aleyasin et al., 2018; Golden et al., 2019) – we speculate that dopamine receptor D1 MSN cells are involved in prosocial intent. This hypothesis would require further testing and will be an interesting area for future study.”

8) The NAc has been the focus of intense scrutiny across numerous social behaviors, ranging from stress-related social anhedonia to sexual motivation to aggression motivation. The discussion would notably strengthen by relating the current results within the context of previously identified roles for the NAc in other forms of social motivation behaviors.

We hope the new paragraph in the discussion (quoted above) provides a resolution of this concern.

5) There is no Figure 2G, referenced in line 156.

This typo has been fixed, thank you!

[Editors' note: further revisions were suggested prior to acceptance, as described below.]

The manuscript has been improved but there are some remaining issues that need to be addressed, as outlined below:The concerns regarding the original submission have been addressed. The revised manuscript provides novel insight regarding the role of the nucleus accumbens and surrounding circuitry in prosociality, and will therefore be of broad interest to the scientific community. The reviewers have a few remaining concerns regarding the Figures that need to be addressed:1) The representative cFos image shown in Figure 2B should be replaced with a cFos image from the nucleus accumbens, as this is the brain region that was used for recording.2) To better illustrate the co-labels with cFos in the merged image in Figure 5D, the authors should consider pseudo-coloring the flurogold label.

We have modified the two figures as requested.